# How Private is Your Attention?
# Bridging Privacy with In-Context Learning

**Soham Bonnerjee**
*sohambonnerjee@uchicago.edu*
*Department of Statistics*
*University of Chicago*

**Kingsley Yeon**
*yeon@uchicago.edu*
*Department of Statistics*
*University of Chicago*

**Anna Asch**
*aasch@uchicago.edu*
*Department of Statistics*
*University of Chicago*

**Sagnik Nandy**
*nandy.15@osu.edu*
*Department of Statistics*
*Ohio State University*

**Promit Ghosal**
*promit@uchicago.edu*
*Department of Statistics*
*University of Chicago*

**Reviewed on OpenReview:** *https://openreview.net/forum?id=M2qsrIbaOL*

## Abstract

In-context learning (ICL)—the ability of transformer-based models to perform new tasks from examples provided at inference time—has emerged as a hallmark of modern language models. While recent works have investigated the mechanisms underlying ICL, its feasibility under formal privacy constraints remains largely unexplored. In this paper, we propose a differentially private pretraining algorithm for linear attention heads and present the first theoretical analysis of the privacy–accuracy trade-off for ICL in linear regression. Our results characterize the fundamental tension between optimization and privacy-induced noise, formally capturing behaviors observed in private training via iterative methods. Additionally, we show that our method is robust to adversarial perturbations of training prompts, unlike standard ridge regression. All theoretical findings are supported by extensive simulations across diverse settings.

## 1 Introduction

Attention-based models, particularly large language models (LLMs), have demonstrated remarkable capabilities in performing *in-context learning* (Brown et al., 2020; Lieber et al., 2021; Rae et al., 2021; Black et al., 2022; Bubeck et al., 2023). This paradigm has transformed human-AI interaction, enabling AI models to tackle complex tasks without explicit parameter updates. A growing body of theoretical work has aimed to explain this emergent behavior (Dong et al., 2022; Akyürek et al., 2022; Garg et al., 2022; Wang et al.,

2023; Xie et al., 2022), often using simplified settings. These studies suggest that transformers can implicitly infer patterns or rules from training examples in the prompt and apply them to new, related inputs during inference.

The growing use of LLM-based agents in sensitive domains such as medicine (Li et al., 2025; Dennstädt et al., 2025) and psychology (Ke et al., 2024) underscores the urgent need for robust privacy safeguards. In particular, model providers must prevent adversaries from extracting information about sensitive training prompts by inspecting the released model weights. Such privacy leakage risk was highlighted by recent work demonstrating that LLMs can memorize and reveal specific training examples when prompted adversarially (Carlini et al., 2021; 2022; Tirumala et al., 2022). A principled approach to mitigating such leakage is *differential privacy* (DP) (Dwork et al., 2006), which ensures that an algorithm's output remains nearly unchanged when a single training prompt for an in-context learning task is modified. This is typically achieved by injecting calibrated noise to limit individual influence.

However, integrating privacy-preserving mechanisms into the pretraining process of a transformer inevitably degrades the downstream performance of in-context learning on test prompts. This trade-off motivates a rigorous study of the cost of privacy of *in-context differentially-private* algorithms: what additional error is incurred at test time?

## 1.1 Main Results

We study the effect of differentially-private pretraining on in-context learning (ICL) for linear regression, where each data point is a noisy linear response to input features. The model is trained on $N$ prompts, each containing $L$ feature-response pairs sampled from a noisy linear model, and optimized to minimize squared prediction error on the query token. We propose a differentially-private pretraining algorithm for a linear attention head that performs ICL—predicting the response for a query input by attending to a sequence of labeled input-output examples. Our goal is to ensure that adversarial inspection of the released model parameters (or the predicted output) does not reveal information about the inclusion of any specific prompt in the training examples.

It is important to emphasize that our problem is quite different from classical regression. In traditional linear regression, the model learns a single weight vector from the training data and applies it to new inputs. In contrast, in our setting, the model is trained to predict the labels for a new input query given a context of labeled examples from a task-specific model parameter, drawn anew for each episode. The model does not learn this model parameter explicitly, but rather uses a shared decoder (linear attention head) to infer patterns from the context, similar to how transformers adapt to prompts without updating parameters.

To enforce privacy in our pre-training scheme, we apply the Gaussian mechanism with gradient clipping followed by additive noise—commonly used in private empirical risk minimization (Dwork et al., 2006; Chaudhuri et al., 2011; Abadi et al., 2016; Cai et al., 2021). Our method, `NoisyHead` (Algorithm 1, Section 3), formalizes this approach.

We define the *cost of privacy* as the difference, between attention heads trained with and without privacy constraints, in average prediction error of the response to a query token from a held-out test prompt. Our main theoretical result characterizes how the *cost of privacy* scales with the number of training prompts $N$, the prompt length $L$, the token dimension $D$, and the privacy parameters $(\varepsilon, \delta)$. We state it informally below:

**Theorem 1.1** (Informal). *In the low dimensional regime, when $L$ and $\sqrt{N}$ are asymptotically of same order and $D = O(1)$, the cost of privacy satisfies*

$$Cost\ of\ Privacy \lesssim \frac{1}{N^{3/2} L^2} \frac{\log(1/\delta)}{\varepsilon^2}.$$

*In the high dimensional regime, when $N/D^2 = O(1)$ and $L/D = O(1)$, the cost of privacy scales as*

$$Cost\ of\ Privacy \lesssim \frac{D^2}{N^2 L^2} \frac{\log(1/\delta)}{\varepsilon^2},$$

*up to* polylog *factors.*

A formal version of this result is presented in Theorem 4.2, followed by a detailed discussion of its implications. The theorem highlights that the cost of privacy exhibits fundamentally different behavior in the low- and high-dimensional regimes. In the *low-dimensional* setting, the minimax cost of privacy for learning a linear model from $L$ labeled data points is known to scale as $(\varepsilon L)^{-2} \cdot \log(1/\delta)$, as established in Cai et al. (2021). The result above shows that leveraging contextual data reduces this cost to $N^{-3/2}(\varepsilon L)^{-2} \cdot \log(1/\delta)$. However, because test-time prediction requires learning an unseen coefficient vector $w$, we do not achieve the rate $N^{-2}(\varepsilon L)^{-2} \cdot \log(1/\delta)$, which would be expected if the coefficient was identical across all training and test prompts. In contrast, in the *high-dimensional* regime, where the feature dimension scales with the number of prompts $N$, we incur an additional multiplicative factor of $\sqrt{N}$ in the denominator due to the increased complexity of the learning problem.

We also show that our private pretraining procedure is more robust to adversarial perturbations of training prompts than its non-private counterpart. When a fraction of prompts are corrupted, the prediction risk on test instances remains significantly more stable under our method — a property especially relevant given recent concerns about adversarial attacks in LLMs (Anwar et al., 2024).

Our key contributions are as follows:

(**1**) We propose a differentially-private pretraining algorithm (`NoisyHead`) based on the *Gaussian mechanism* for training linear attention heads to perform in-context learning in linear regression (see Algorithm 1). Our method is motivated by the differentially-private stochastic gradient descent algorithm (Abadi et al., 2016), containing a tuned noise-injection at the gradient steps.

(**2**) We provide a detailed theoretical analysis of the excess risk incurred by enforcing differential privacy during pretraining in Theorem 4.1. In particular, it characterizes the privacy–utility trade-off, quantifying the impact of privacy constraints on the prediction error of `NoisyHead` across any number of iterations $T$ of the algorithm. This trade-off exhibits dichotomous behavior depending on how the feature dimension $D$ scales with the number of training samples $N$. We identify two distinct regimes: one where $D = O(\log N)$ and another where $N/D^2 = O(1)$. These lead to qualitatively different error decay rates with respect to $N$, $L$, and $D$, as formalized in Theorem 4.2. In the over-parametrized setting when $N, L^2, D^2$ are asymptotically of the same order, we show that there is a delicate interplay between the number of training iterations and the generalization error on unseen prompts. Due to the injection of noise at each iteration, longer training can degrade generalization, necessitating careful selection of the number of optimization steps. This highlights the importance of "early stopping" for the algorithm. See Proposition 4.1 and the following remark for related discussion.

(**4**) We establish that `NoisyHead` exhibits a notable robustness property under adversarial perturbations to the training data, particularly during the pretraining stage. Compared to the baseline method proposed in Lu et al. (2024), our approach shows significantly less degradation in generalization error in the presence of such perturbations. In the baseline setting, where the linear attention module is pretrained using ridge regression, even moderately large perturbations can induce a distributional shift in the training data, leading to inaccurate estimation of model weights and consequently poor generalization. In contrast, `NoisyHead` incorporates a truncation mechanism that clips responses, predictors, and weights within prescribed compact sets. This simple yet effective step restricts the influence of corrupted or outlying data points, enhancing robustness to adversarial noise introduced during training. Theoretical support for this robustness is provided in Theorem 5.1.

(**5**) We conduct a comprehensive empirical study to validate the theoretical predictions of our analysis. In both low- and high-dimensional regimes (Section 6.1), we demonstrate that the excess prediction risk of `NoisyHead` decays with increasing sample size and privacy parameter, consistent with the rates derived in Theorem 4.2. Moreover, in the overparameterized regime (Section 6.2), our experiments reveal a distinct phase transition in the generalization error: initially decreasing due to optimization, but eventually increasing due to cumulative noise from differential privacy. This phenomenon, visualized in Figure 3, substantiates the theoretical trade-off outlined in Proposition 4.1 and underscores the critical role of early stopping. Finally, robustness experiments (Section 6.3) confirm that `NoisyHead` maintains stable performance under adversarial perturbations, while ridge-based pretraining degrades significantly. These results highlight the practical utility of our method and affirm the relevance of our theoretical contributions in realistic settings.

In this paper, we focus on linear regression and linear attention in contrast to a general transformer-based architecture because it provides a clean and analyzable framework to study the impact of private pre-training on in-context learning performance of attention-based models. While the idea underlying the design of `NoisyHead` can also be extended to privately pre-train general transformers for in-context learning tasks in more complicated settings, it is quite hard to explicitly characterize the impact of enforcing privacy on the test time performance. This toy model provides an avenue to explore such a privacy-utility trade-off in a mathematically quantifiable way.

It is worthwhile to mention that some recent works have also explored differentially private fine-tuning on complicated models such as GPT-2 or ViT (Yu et al., 2023; Ding et al., 2024; Oh et al., 2024), which mainly apply DP-SGD to fine-tune such models. Moreover `NoisyHead` structurally, resembles a carefully designed DP-GD pipeline that utilizes the structure of the specific in-context learning problem and the model architecture to carefully calibrate the amount of noise infused in the pipeline. While both our pipeline and the standard DP-SGD pipeline involve clipping, projection, and noise injection, they differ significantly in structure and intent. In particular, if applied naively, DP-SGD would treat the entire sequence of $L$ context-label pairs as private and apply noise accordingly. This would cause the privacy cost to compound across the entire episode, leading to substantial degradation in accuracy. In contrast, `NoisyHead` leverages the structure of in-context learning by injecting noise only at the level of the final gradient, using carefully designed clipped sufficient statistics that summarize each episode in a privacy-preserving way. This includes clipping of the response variables $y_{k,i}$ before computing sufficient statistics-a step not typically used in standard DP-SGD but essential for both privacy guarantees and statistical accuracy, particularly with long contexts. Furthermore, our clipping constants are derived using the model assumptions on the prompt level features and responses using appropriate concentration inequalities. While this derivation is inspired by analogous analyses in linear regression settings Cai et al. (2021), the corresponding calculations for the in-context learning problem are significantly more involved and, to the best of our knowledge, have not been explored previously. Finally, DP-SGD is mainly designed for classical supervised learning, where model parameters are updated with each example. In contrast, `NoisyHead` generalizes across tasks presented at test time and injects noise only at the head level, using aggregated, privatized task-level statistics. Finally, despite showing promising empirical performance, DP-SGD-based pre-training of attention modules often lacks theoretical guidance on how privacy cost scales with model structure, sample size, or representation geometry. In contrast, we precisely characterize the amount of extra price paid in the prediction loss due to enforcing privacy in the training pipeline at the user-specified threshold. Our theoretical insights provide a rigorous foundation that could inform the design of future DP algorithms for large-scale models with a quantifiable characterization of the cost of privacy in the in-context learning performance.

## 1.2 Related literature and notations

Since its introduction by Dwork et al. (2006), differential privacy has become a cornerstone of privacy-preserving machine learning, inspiring a wide range of algorithms across classical and deep learning tasks (Cai et al., 2021; Wang and Xu, 2019; Gu et al., 2024; Jain and Thakurta, 2013; Ni et al., 2016; Ji et al., 2019; Abadi et al., 2016; Feldman et al., 2018). However, most of these algorithms are primarily concerned with the development of private iterative optimization schemes in connection to training learning models for standard supervised or unsupervised problems. While privacy utility guarantees are known for a subset of these methods, they do not readily extend to the characterization of analogous guarantees for the privacy-utility tradeoff for in-context learning problems, where the inference-time prediction task is related to—but not identical to—the problem instances observed in the training prompts.

A number of recent papers have tried to understand the mechanisms to ensure differentially-private pretraining for transformers and evaluated the privacy properties of language models. This includes differentially private decoding of language models Majmudar et al. (2022), private fine-tuning of language models Yu et al. (2023), and private learning properties of large language models (Li et al., 2022; Hoory et al., 2021; Anil et al., 2021) like GPT or BERT. Furthermore, McMahan et al. (2018) studied private learning of recurrent language models, and Beigi et al. (2019) studied private text representation learning.

In parallel, a recent line of work has also explored the in-context learning (ICL) capabilities of transformers in statistical learning tasks, demonstrating that pretraining enables them to emulate diverse learning

algorithms—including ridge regression, generalized linear models, Lasso, and neural networks—purely from contextual examples (Dai et al., 2023), with theoretical insights provided for linear (Zhang et al., 2024; Lu et al., 2024) and softmax attention models (Huang et al., 2023; Yang et al., 2024; Li et al., 2024b; Chen et al., 2024). Despite significant advances in both areas, their intersection remains underexplored. This paper bridges this gap by providing the first rigorous analysis of how enforcing differential privacy during pretraining affects the in-context learning capabilities of attention-based models. To this end, we draw inspiration from the mathematical analysis of the DP-GD algorithm for learning linear regression models in Cai et al. (2021). However, our setting requires substantially more intricate arguments due to the complex functional dependence between prompt features and responses induced by the attention head.

We believe that an analogous analysis could be carried out for a DP-SGD-based pretraining scheme applied to the same linear attention head, and that the corresponding privacy-utility tradeoff could be characterized, but with additional mathematical complexity. Moreover, the privacy accounting in our analysis could be further tightened by adopting more refined noise-injection frameworks, such as Rényi DP or Gaussian DP, in place of standard $(\varepsilon, \delta)$-DP. We leave these extensions to future work.

### 1.2.1 Notation

In this paper, we denote the set $\{1, \ldots, n\}$ by $[n]$. $d$-dimensional Euclidean space is $\mathbb{R}^d$, with $\mathbb{R}^d_{>0}$ the positive orthant. The set of $m \times n$ real matrices is $\mathbb{R}^{m \times n}$, and $\mathbb{S}^{d-1}$ denotes the $d$-dimensional unit sphere. The Frobenius norm of a matrix $A$ is $\|A\|_F$, and $\langle \cdot, \cdot \rangle$ denotes the standard inner product. We write $a_n \lesssim b_n$ if $a_n \leq C b_n$ for some constant $C > 0$, and $a_n \asymp b_n$ if $C_1 b_n \leq a_n \leq C_2 b_n$ for some constants $C_1, C_2 > 0$. We also write $a_n \asymp b_n$ as $a_n = \Theta(b_n)$.

## 2 Problem Formulation

We consider a set-up where we observe a sequence of labeled tokens $\{(y_i, x_i) : i \in \{1, \ldots, L\}\}$, for $x_i \overset{i.i.d}{\sim} \mathcal{U}(\mathbb{S}^{D-1})$ and $y_i = w^\top x_i + \epsilon_i$, with $w \sim \mathcal{N}_D(0, \mathbb{I}_D)$ and $\epsilon_i \overset{i.i.d}{\sim} \mathcal{N}(0, \tau^2)$. Here $\mathcal{U}(\mathbb{S}^{D-1})$ denotes the uniform distribution on the $D$-dimensional hypersphere and $\mathcal{N}_k(\mu, \Sigma)$ denotes the $k$ dimensional normal distribution with mean $\mu$ and covariance $\Sigma$. Note that, it is standard practice to normalize features to lie on the unit sphere, and such an assumption is common in the differential privacy literature; see Cai et al. (2021). For a test token $(y_{L+1}, x_{L+1})$ generated independently from the same distribution as the training tokens, we want to predict $y_{L+1}$ based on $x_{L+1}$. We interpret this Gaussian assumption as referring to noise in the data. While it simplifies analysis, differential privacy still holds under non-Gaussian noise. However, maintaining strong convergence rates may then require adjusted clipping thresholds and sharper concentration bounds—an important but technically involved direction we leave for future work.

This particular setting was used by Zhang et al. (2024) and Lu et al. (2024), both of whom considered the noiseless case of $\tau^2 = 0$. As proposed therein, we embed the prompt as

$$E = \begin{pmatrix} x_1 & x_2 & \cdots & x_L & x_{L+1} \\ y_1 & y_2 & \cdots & y_L & 0 \end{pmatrix} \in \mathbb{R}^{(D+1) \times (L+1)}. \tag{2.1}$$

This matrix is passed through a single linear attention head as follows:

$$f(E; \theta) = E + W^{PV} E \cdot \frac{E^\top W^{KQ} E}{L}, \tag{2.2}$$

where $\theta = (W_{PV}, W_{KQ})$ with $W_{PV} \in \mathbb{R}^{(D+1) \times (D+1)}$ and $W_{KQ} \in \mathbb{R}^{(D+1) \times (D+1)}$. The prediction of the query response is given by the $(D+1, L+1)$-th entry of $f(E; \theta)$; that is, $\widehat{y}_{L+1}(E) = (f(E; \theta))_{(D+1, L+1)}$. We aim to learn the parameters of the model $f(E; \theta)$ by pretraining the model based on $N$ training prompts $\{(y_{k,1}, x_{k,1}), \ldots, (y_{k,L}, x_{k,L}), (y_{k,L+1}, x_{k,L+1})\}_{k=1}^N$, where the $L+1$-th token is the query token. Putting the prompts into matrices $E_1, \ldots, E_N$, we have

$$E_k := \begin{pmatrix} x_{k,1} & x_{k,2} & \cdots & x_{k,L} & x_{k,L+1} \\ y_{k,1} & y_{k,2} & \cdots & y_{k,L} & 0 \end{pmatrix} \in \mathbb{R}^{(D+1) \times (L+1)}.$$

Now we minimize the standard loss function $\mathcal{L}(\theta) = \frac{1}{2N}\sum_{i=1}^{N}(\widehat{y}_{L+1}(E_k) - y_{k,L+1})^2$. The predictor $(f(E;\theta))_{(D+1,L+1)}$ can be simplified by linear algebra to

$$\hat{y}_{L+1} := [(f(E;\theta)_{[D+1,L+1]}] = \begin{bmatrix} \left(w_{21}^{PV}\right)^{\top} & w_{22}^{PV} \end{bmatrix} \left(\frac{EE^{\top}}{L}\right) \begin{bmatrix} W_{11}^{KQ} \\ \left(w_{21}^{KQ}\right)^{\top} \end{bmatrix} x_{L+1}, \tag{2.3}$$

where we have used the matrices $W^{PV}$ and $W^{KQ}$, partitioned as follows:

$$W^{PV} = \begin{bmatrix} W_{11}^{PV} & w_{12}^{PV} \\ \left(w_{12}^{PV}\right)^{\top} & w_{22}^{PV} \end{bmatrix}, \quad W^{KQ} = \begin{bmatrix} W_{11}^{KQ} & w_{12}^{KQ} \\ \left(w_{12}^{KQ}\right)^{\top} & w_{22}^{KQ} \end{bmatrix},$$

with $W_{11}^{PV}, W_{11}^{KQ} \in \mathbb{R}^{D\times D}$, $w_{21}^{PV}, w_{21}^{KQ} \in \mathbb{R}^{D}$, and $w_{22}^{PV}, w_{22}^{KQ} \in \mathbb{R}$. The quadratic form (2.3) can be expanded to yield

$$\hat{y}_{L+1} = \frac{1}{L}\langle x_{L+1}, Q_W^{(1)} + Q_W^{(2)}\rangle, \tag{2.4}$$

where $Q_W^{(1)} := w_{22}^{PV}W_{11}^{KQ}\sum_{i=1}^{L}y_i x_i + w_{22}^{PV}w_{12}^{KQ}\sum_{i=1}^{L}y_i^2$ and $Q_W^{(2)} := W_{11}^{KQ}\sum_{i=1}^{\ell+1}x_i x_i^{\top}w_{12}^{PV} + w_{12}^{KQ}\sum_{i=1}^{\ell}y_i x_i^{\top}w_{12}^{PV}$. Following Yu et al. (2023) and Zhang et al. (2024), we adopt the assumption that $w_{12}^{KQ} = 0$ and $w_{12}^{PV} = 0$ throughout this paper. This particular choice is also explained in Section A.1. Let us define

$$\Gamma = w_{22}^{PV}W_{11}^{KQ} \in \mathbb{R}^{D\times D}, \quad \text{and} \quad Z = \frac{1}{L}x_{L+1}\sum_{i=1}^{L}y_i x_i^{\top} \in \mathbb{R}^{D\times D}. \tag{2.5}$$

With this definition of $\Gamma$ and $Z$, the predictor $\widehat{y}$ simplifies to the inner product $\widehat{y} = \langle\Gamma, Z\rangle$, and we train the model using the following regularized squared error loss:

$$\mathcal{L}_{\lambda}(\Gamma) := \frac{1}{N}\sum_{i=1}^{N}(y_i - \langle\Gamma, Z_i\rangle)^2 + \lambda\|\Gamma\|_F^2. \tag{2.6}$$

The solution to this optimization problem is denoted by $\Gamma^{\star} \in \mathbb{R}^{D\times D}$, whose vectorized form is given by

$$\text{vec}(\Gamma^{\star}; E_1, \ldots, E_N) = \left(\lambda NI + \sum_{k=1}^{N}\text{vec}(Z_k)\text{vec}(Z_k)^{\top}\right)^{-1}\sum_{k=1}^{N}y_{k,L+1}\text{vec}(Z_k). \tag{2.7}$$

---

**Algorithm 1** In-Context Differentially private pretraining of linear attention head (`NoisyHead`)

---

**Input:** Training prompts $(E_k)_{k\in[N]} \in \mathbb{R}^{(D+1)\times(L+1)}$; noise scale $\sigma$; privacy parameters $\varepsilon, \delta$; clipping parameter $\mathcal{C} \geq 0$; projection parameters $R, G \geq 0$; regularization parameter $\lambda := \lambda(n,d) \geq c > 0$; number of iterations $T$; step-size $\eta_0$; and initialization $\Gamma^0 \in \mathbb{R}^{D\times D}$ with $\|\Gamma^0\|_F \leq R$.

- For $k \in [N]$, $\widetilde{Z}_k := \Pi_G\left(L^{-1}x_{k,L+1}\sum_{i=1}^{L}\texttt{clip}_{\mathcal{C}}(y_{k,i})x_{k,i}^{\top}\right)$.

- For $t$ in $0, 1, \ldots, T-1$:

  - Generate $z_t \in \mathbb{R}^{D\times D}$ such that $\text{vec}(z_t) \sim \mathcal{N}_{D^2}\left(0, 2\eta_0^2\frac{T^2\sigma^2}{\varepsilon^2 N^2}\log\frac{1.25T}{\delta}\mathbb{I}_{D^2}\right)$.

  - Do $\Gamma^{t+1} = \Pi_R\left((1-2\lambda\eta_0)\Gamma^t - \eta_0 N^{-1}\sum_{k=1}^{N}\left(\langle\Gamma^t, \widetilde{Z}_k\rangle - \texttt{clip}(y_{k,L+1})\right)\widetilde{Z}_k + z_t\right)$.

  **Output:** $\hat{\Gamma} := \Gamma^T$.

---

# 3 Differentially Private Pretraining

In this section, we present our differentially-private pretraining program of a linear attention network. Before proceeding to the main algorithm, we recall the definition of differential privacy.

**Definition 3.1.** *A randomized algorithm $\mathcal{M}(\cdot)$ over a set of prompts is said to be in-context $(\varepsilon, \delta)$-differentially private if for any two sequences of prompts $\mathcal{D} = (E_1, \ldots, E_N)$ and $\mathcal{D}' = (E'_1, \ldots, E'_N)$ differing in at most one entry, and for all measurable subsets $\mathcal{W}$ of outputs,*

$$\mathbb{P}[\mathcal{M}(\mathcal{D}) \in \mathcal{W}] \leq e^{\varepsilon} \mathbb{P}[\mathcal{M}(\mathcal{D}') \in \mathcal{W}] + \delta.$$

The probability is taken over the internal randomness of the mechanism $\mathcal{M}$, while the prompt sequences $\mathcal{D}$ and $\mathcal{D}'$ are treated as fixed. This definition ensures that the inclusion or exclusion of any individual data point has a limited effect on the algorithm's output, thereby preserving privacy. A standard approach to enforce DP in the iterative training of machine learning models (e.g. gradient descent) is to inject noise at each update step. The cumulative effect of this noise is carefully calibrated to satisfy user-specified $(\varepsilon, \delta)$-differential privacy guarantees but minimize degradation in model performance. This technique, introduced as *differentially private stochastic gradient descent*, has been echoed in recent works (Abadi et al., 2016; Cai et al., 2021; Zhang et al., 2021; Gopi et al., 2021; Majmudar et al., 2022; Bombari and Mondelli, 2025). In what follows, we improvise the aforementioned differentially-private training strategy while using the gradient descent to minimize the regularized loss $\mathcal{L}_{\lambda}(\Gamma)$ over a sequence of prompts:

$$\Gamma^{t+1} = (1 - 2\lambda\eta_0)\Gamma^t - \frac{\eta_0}{N} \sum_{k=1}^{N} \left( \langle \Gamma^t, Z_k \rangle - y_{k,L+1} \right) Z_k,$$

where $\eta_0$ is the learning rate, and $\lambda$ is the regularization parameter.

To ensure privacy, we inject carefully calibrated Gaussian noise into each update step. The variance of this noise is set proportional to the $\ell_2$-*sensitivity* of the update, which measures the maximum change in the update (in Frobenius norm) resulting from the change of a single training example. Formally, the $\ell_2$-sensitivity at iteration $t$ is defined as:

$$\Delta(\hat{\Gamma}) = \left\| \hat{\Gamma}(E_1, \ldots, E_N) - \hat{\Gamma}(E'_1, \ldots, E'_N) \right\|_F, \tag{3.1}$$

where the datasets $(E_1, \ldots, E_N)$ and $(E'_1, \ldots, E'_N)$ differ in exactly one training prompt. Intuitively, privacy is preserved because an adversary observing the output of the algorithm (i.e., the final parameters) cannot reliably distinguish whether a change in the result is due to the presence or absence of a particular training prompt or due to the added random noise. However, in the problem setup considered in this paper, the $\ell_2$-sensitivity of the gradient updates may not be uniformly bounded across all possible sequences of training prompts due to the unbounded nature of the weights $w$ and the noise $\epsilon$. To mitigate this, we clip the responses $y_k$ and project the gradient updates $\Gamma^t$ onto compact sets, which automatically bounds the sensitivity of the parameters at a fixed dictated by the diameter of the set. With these modifications, our differentially-private pretraining algorithm is presented in Algorithm 1, where the clipping and projection operators are defined as follows:

$$\texttt{clip}_{\mathcal{C}}(x) := \arg \min_{y \in [-\mathcal{C}, \mathcal{C}]} \|x - y\|_2, \quad \Pi_R(X) := \arg \min_{\substack{Y \in \mathbb{R}^{(D+1) \times (D+1)} \\ \|Y\|_F \leq R}} \|X - Y\|_F.$$

We remark that response clipping is a well-established technique to ensure differential privacy; it has been widely used in both theoretical and applied works (e.g., Ding et al. (2024)). In particular, such mechanisms ensures differential privacy, as formalized by Theorem 3.2.

**Theorem 3.2.** *Given the set of hyperparameters $(\mathcal{C}, R, G) \in \mathbb{R}^3_{>0}$, Algorithm 1 is $(\varepsilon, \delta)$-differentially private if the noise scale $\sigma \geq 2G(\mathcal{C} + RG)$.*

Theorem 3.2 (which we prove in Section A.2) hints at the minimum amount of noise to be injected in the gradient descent step to achieve differential privacy. In particular, the amount of noise depends crucially

on the projection parameters $\mathcal{C}$ and $R$. On the other hand, the higher the noise variance $\sigma^2$, the more we expect the predictive performance of the differentially-private estimate $\hat{\Gamma}$ to degrade compared to the ridge estimate $\Gamma^\star$ (as defined in (2.6)). However, it can still be argued that performing an appropriate number of iterations, governed by an "early stopping criterion", can improve accuracy. In fact, the additional error from noise injection can be made much smaller than the overall gradient descent error by properly tuning the hyper-parameters. This angle is explored in detail in Section 4. We conclude this section by discussing two key aspects of our algorithms.

*Remark* 3.1. An alternative approach for estimating $\Gamma$ in this set-up can be computing the optimizer in (2.6) and adding calibrated Gaussian noise to the solution to obtain an $(\varepsilon, \delta)$-DP estimator. We provide a detailed discussion on this approach in Section B.

## 4 Cost of In-Context Differential Privacy

In this section, we rigorously characterize the additional error incurred due to enforcing privacy constraints in Algorithm 1. Let $E^{\texttt{test}}$ be a test prompt and $y_{L+1}^{\texttt{test}}$ be the corresponding query response. Let us consider the prediction error in the test prompt given by

$$\mathcal{L}_{\texttt{test}}(\Gamma) = (y_{L+1}^{\texttt{test}} - \langle \Gamma, Z(E^{\texttt{test}}) \rangle)^2,$$

where $Z(E^{\texttt{test}})$ is constructed from $E^{\texttt{test}}$ as described in (2.5). We bound $\mathcal{L}_{\texttt{test}}(\Gamma)$ by the following two types of error terms:

$$\mathcal{L}_{\texttt{test}}(\hat{\Gamma}) \leq 2\mathcal{L}_{\texttt{test}}(\Gamma^\star) + 2(\langle \hat{\Gamma}, Z(E^{\texttt{test}}) \rangle - \langle \Gamma^\star, Z(E^{\texttt{test}}) \rangle)^2.$$

While $\mathcal{L}_{\texttt{test}}(\Gamma^\star)$ is the prediction error of the non-private procedure, the extra error is proportional to $(\langle \hat{\Gamma}, Z(E^{\texttt{test}}) \rangle - \langle \Gamma^\star, Z(E^{\texttt{test}}) \rangle)^2$. The following theorem characterizes this extra error.

**Theorem 4.1.** *Consider the pretrained weights $\hat{\Gamma}$ generated by running `NoisyHead` (Algorithm 1) on prompt set $(E_1, \ldots, E_N)$, ensuring $(\varepsilon, \delta)$ differential privacy for $T$ iterations with a fixed stepsize $\eta_0 \in \left( \frac{\lambda}{c(2\lambda + G^2)^2}, \frac{\lambda}{(2\lambda + G^2)^2} \right)$ for some large $c > 1$, and $\Gamma^\star$ generated by solving the ridge regression described in (2.6). If the clipping and projection parameters are set as:*

$$\nu = 1 + \tau^2, \quad \mathcal{C} = \sqrt{2\nu \log(NL/\kappa)}, \quad G = \frac{\mathcal{C}}{\sqrt{L}} \left( 1 + \frac{(\log(N/\kappa))^{1/2}}{D} \right),$$

$$G_0 = \frac{\mathcal{C}}{\sqrt{L}} \left( 1 + \frac{(\log(1/\kappa))^{1/2}}{D} \right), \quad \text{and } R \asymp \lambda^{-1} \mathcal{C}^2 \sqrt{\frac{N}{L}} \left( 1 + \frac{(\log(1/\kappa))^{1/2}}{D} \right), \tag{4.1}$$

*then for a test prompt $E$ independent of $(E_k)_{k \in [N]}$,*

$$(\langle \hat{\Gamma}, Z \rangle - \langle \Gamma^\star, Z \rangle)^2 \leq G_0^2 \left( (1 - \eta_0 \lambda)^T R^2 + \sigma^2 \eta_0^2 D \frac{T^2 \log(2T/\delta)}{N^2 \varepsilon^2} \right). \tag{4.2}$$

*with probability greater than $1 - c_1 \exp(-c_2 D) - 4\kappa$, where $Z$ is formed via $E$ as in (2.5).*

The above theorem is proved in Section A.3.

*Remark* 4.1. The "*cost of privacy*" on the right hand side of (4.2) naturally decomposes into two components. The first arises from the optimization error of gradient descent, hereby referred to as the "*cost of descent*", and is given by $(1 - \eta_0 \lambda)^T$, where $\eta_0$ is the step size, $\lambda$ is the strong convexity parameter, and $T$ is the number of gradient steps. The second component stems from the noise injected at each iteration to ensure DP, and takes the form $\sigma^2 \eta_0^2 D \cdot \frac{T^2 \log(2T/\delta)}{N^2 \varepsilon^2}$, where $\sigma^2$ denotes the variance of the added noise, $D$ is the feature dimension, $N$ is the number of training samples, and $(\varepsilon, \delta)$ are the privacy parameters. This term will henceforth be referred to as the "*cost of noise injection*". The trade-off between these two terms plays a crucial role in determining the generalization error. While the optimization error decays exponentially with $T$, the privacy-induced error increases quadratically. Therefore, it is essential to choose an optimal stopping time for the gradient descent iterations. This optimal stopping time depends on the problem hyper-parameters $\eta_0$, $\lambda$, and the feature dimension $D$. In the following theorem (proved in Section A.4), we characterize how the interplay between the dimensionality and the stopping time governs the behavior of the generalization error in different settings of interest.

**Theorem 4.2.** *Assume $N \gtrsim D^2 L^{-2}$, and suppose the noise scale $\sigma$ in Algorithm 1 satisfies $\sigma \asymp 2G(\mathcal{C} + RG)$ where $(\mathcal{C}, R, G)$ is the set of hyper-parameters. Then, under the assumptions and hyper-parameter specifications of Theorem 4.1, the following assertions hold:*

*(i) (Low-dimensional setting) If $\kappa \gtrsim \exp(-D^2)$, and $D^2 \lesssim \log(NL)$, then, after $T = \frac{\log(N^2 L D^3)}{\log((1-\eta_0\lambda)^{-1})}$ many iterations of Algorithm 1 with $\eta_0 \asymp \lambda \asymp 1$ such that $\eta_0 \lambda \in (0, 1)$, the cost of privacy of $\hat{\Gamma}(= \Gamma^T)$ behaves as follows:*

$$(\langle \hat{\Gamma}, Z \rangle - \langle \Gamma^\star, Z \rangle)^2 \lesssim \nu^5 \frac{\log^{10}(NL)}{NL^3} \left(1 + \frac{\log(1/\delta)}{\varepsilon^2}\right), \tag{4.3}$$

*with probability at least $1 - c_1 \exp(-c_2 D) - 4\kappa$.*

*(ii) (High-dimensional setting) If $\kappa \gtrsim (NL)^{-1}$, and $D^2 \gtrsim \log(NL)$, then for some $r$ (possibly depending on $N, L$ and $D$), let $T = \frac{\log r}{\log((1-\eta_0\lambda)^{-1})}$. If $\eta_0 < \frac{\lambda}{(2\lambda+G^2)^2}$, then*

$$\begin{aligned} &(\langle \hat{\Gamma}, Z \rangle - \langle \Gamma^\star, Z \rangle)^2 \\ &\lesssim \frac{N\nu^5 \log^3(NL)}{L^2 \lambda^2 r}\left(1 + \frac{D \, r \log^3 r}{N^3}\left(1 + \log^2(NL)\frac{N}{L^2\lambda^2}\right)\frac{\log(1/\delta)}{\varepsilon^2}\right), \end{aligned} \tag{4.4}$$

*with probability at least $1 - c_1 \exp(-c_2 D) - 4\kappa$.*

*Remark* 4.2. In the low-dimensional setting with a specific choice of $T$, the "cost of gradient descent" is negligible, and the "cost of noise injection" becomes the dominant contributor to the overall "cost of privacy". Notably, the restriction on $D$ renders it irrelevant in determining the cost of privacy in this regime. On the other hand, among the many possible high-dimensional scenarios, a particularly interesting case is the over-parameterized regime where $N \asymp L^2 \asymp D^2$.

*Remark* 4.3. Although the objective function in (2.6) formally resembles ridge regression, the underlying feature construction is fundamentally different. In our setting, the features are not directly observed; rather, they are constructed through nontrivial transformations of both the covariates and responses within each training prompt, mediated by the attention mechanism in (2.4). As a result, the subsequent analysis of prediction risk departs substantially from that of classical differentially private ridge regression.

In standard private ridge regression, prediction accuracy is primarily governed by the sample size $N$ and the ambient dimension $D$. In contrast, the in-context learning (ICL) setting introduces three interacting parameters: the prompt length $L$, the token dimension $D$, and the number of training prompts $N$. This additional structural complexity leads to statistical rates and privacy-utility trade-offs that are qualitatively different from those in classical DP ridge regression, as well as from earlier related works such as Wu et al. (2017).

**Proposition 4.1.** *If $N \asymp L^2 \asymp D^2$, then with $\lambda \asymp \frac{N}{D}$, it holds that*

$$\left(\langle \hat{\Gamma}, Z \rangle - \langle \Gamma^\star, Z \rangle\right)^2 \lesssim \nu^5 \log^3(NL)\frac{D^2}{NL^2 r}\left(1 + r\log^3 r \cdot \frac{D}{N^3} \cdot \frac{\log(1/\delta)}{\varepsilon^2}\right), \tag{4.5}$$

*with probability at least $1 - c_1 \exp(-c_2 D) - 4\kappa$. In particular, when $r \asymp N$, or, equivalently, $T \asymp \log N$, it holds that*

$$\left(\langle \hat{\Gamma}, Z \rangle - \langle \Gamma^\star, Z \rangle\right)^2 \lesssim \nu^5 \log^3(NL)\frac{D^2}{N^2 L^2} \tag{4.6}$$

*with probability at least $1 - c_1 \exp(-c_2 D) - 4\kappa$.*

This result is proved in Section A.5. The choice of $\lambda$ in Proposition 4.1 is standard and also appears in the ridge regression analysis of Lu et al. (2024). Equation (4.5) highlights the trade-off between the two components of the cost of privacy, as previously discussed in Remark 4.1. For fixed values of $N$, $L$, and $D$, the test risk of the estimates generated by `NoisyHead` decreases with the number of iterations $T$ at a rate of $\Theta(e^{-T})$ whereas the test error increases at a rate of $\Theta(T^3)$. As a result, in this high-dimensional regime, the *optimal stopping point* for pretraining is $T = \Theta(\log N)$ iterations. This phenomena is explored numerically in Section 6.1.

## 5 Robustness properties of `NoisyHead`

In this section, we demonstrate that `NoisyHead` is inherently robust to adversarial perturbations to the training data. Specifically, we show that such perturbations during the pretraining stage affect the generalization error of our method significantly less than the baseline approach proposed in Lu et al. (2024).

Consider a set of training prompts $E_1, \ldots, E_N$, and suppose a malicious attacker aims to degrade performance on an independent test prompt $E$ by perturbing the training data, thereby inducing inaccurate estimation of the weights in the attention module. To disrupt the training process, the attacker selects a prompt uniformly at random from the training set, say $E_i$, and replaces it with a perturbed version,

$$E_{\text{bad},i}(\mu, \alpha) = \begin{pmatrix} x'_{i,1} & x'_{i,2} & \cdots & x'_{i,L} & x'_{i,L+1} \\ y'_{i,1} & y'_{i,2} & \cdots & y'_{i,L} & 0 \end{pmatrix} \in \mathbb{R}^{(D+1)\times(L+1)}, \tag{5.1}$$

where the perturbed components are given by $x'_{i,k} = x_{i,k} + \mu$ for all $k \in [L+1]$ and $y'_{i,\ell} = y_{i,\ell} + \alpha$ for all $\ell \in [L]$. Let the parameter trained by the `NoisyHead` algorithm acting on the perturbed set of prompts $(E_1, \ldots, E_{i-1}, E_{\text{bad},i}, E_{i+1}, \ldots, E_N)$ be $\hat{\Gamma}_{\text{bad}}$. Correspondingly, let the parameter trained on the original, unperturbed prompts be $\hat{\Gamma}$. Let the ridge regression solutions of (2.7) on the "perturbed" and "original" set of prompts, be denoted by $\Gamma^\star_{\text{bad}}$ and $\Gamma^\star$, respectively. Then the following theorem characterizes the robustness properties of the estimates generated by `NoisyHead`.

**Theorem 5.1.** *Consider the `NoisyHead` algorithm with the hyper-parameter specifications as in Theorem 4.1. Further, consider an adversarial prompt perturbation as in (5.1), with $\mu, \alpha$ satisfying*

$$\alpha^2 \mu^4 \leq c_u N L \lambda \quad and \quad \alpha^2 \mu^2 \geq c_\ell \mathcal{C}^2 L^{-1/2}(1 \vee \lambda N R^2 L^{-1/2}), \tag{5.2}$$

*for large enough constant $c_u > 0$ and small enough constant $c_\ell > 0$. If $\kappa > Ne^{-D^2}$ and $\lambda > \mathcal{C}^2 L^{-1}$, then for an "unperturbed" test prompt $E$ and the corresponding $Z$ from (2.5), it holds that*

$$(\langle \hat{\Gamma}, Z \rangle - \langle \hat{\Gamma}_{bad}, Z \rangle)^2 \lesssim \frac{N}{L^2} \log^2(NL/\kappa) < \frac{\alpha^2 \mu^2}{N\lambda} \leq (\langle \Gamma^\star, Z \rangle - \langle \Gamma^\star_{bad}, Z \rangle)^2, \tag{5.3}$$

*with probability at least $1 - c_1 \exp(-c_2 D) - 5\kappa$ for constants $c_1, c_2 > 0$.*

*Remark* 5.1. Theorem 5.1 (proved in Section A.6) shows that under bounded perturbations, pretraining with `NoisyHead` yields generalization error closer to that from the unperturbed setup compared to what is acieved by the ridge regression. If $\lambda \asymp 1$ and $D^2 \gtrsim \log N$, the bounds in (5.2) simplify to $\frac{N^2}{L^2} \lesssim \alpha^2 \mu^2 \leq \alpha^2 \mu^4 \lesssim NL$. In the regime $\frac{N}{L^2} \log^2(NL) \to 0$, an adversary can choose $\alpha, \mu$ such that $\alpha^2 \mu^2 \to \infty$ while still satisfying $\alpha^2 \mu^4 \lesssim NL$, leading to $(\langle \Gamma^\star, Z \rangle - \langle \Gamma^\star_{bad}, Z \rangle)^2 \xrightarrow{\mathbb{P}} \infty$. In contrast, `NoisyHead` ensures $(\langle \hat{\Gamma}, Z \rangle - \langle \hat{\Gamma}_{\text{bad}}, Z \rangle)^2 \xrightarrow{\mathbb{P}} 0$ even under such adversarial conditions, as confirmed by experiments in Section 6.3.

## 6 Numerical experiments

We evaluate the empirical behavior of the `NoisyHead` algorithm. Section 6.1 examines how prediction risk changes under different privacy constraint strengths. Section 6.2 explores the trade-off between optimization and noise under different iteration counts. Section 6.3 validates the robustness of `NoisyHead` to adversarial perturbations. All code to reproduce the figures can be found at https://github.com/chebyshevtech/DP1.

### 6.1 Effect of privacy on prediction risk: low- vs. high-dimensional regimes

In this section, we empirically investigate how the level of privacy, parameterized by $\varepsilon$, affects the prediction accuracy of `NoisyHead` through its impact on the excess risk.

**Low-dimensional regime.** We first consider the low-dimensional regime with feature dimension fixed at $D = 5$. Training set sizes are varied over $N \in \{1000, 1500, 2000, 2500, 3000, 3500, 4000\}$, with prompt length set as $L = \lfloor\sqrt{N}\rfloor$, and privacy levels $\varepsilon \in \{0.2, 0.4, 0.6, 0.8, 1.0\}$. The hyperparameters $\mathcal{C}, \mathcal{G}, \mathcal{R}$ are chosen

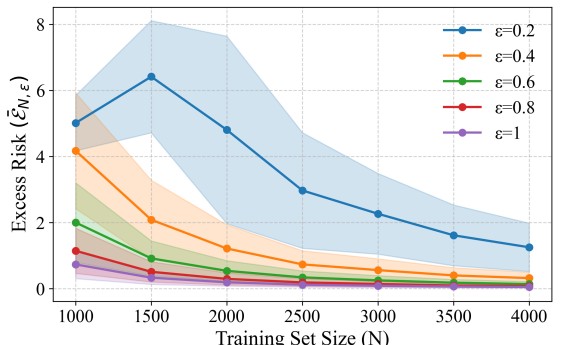 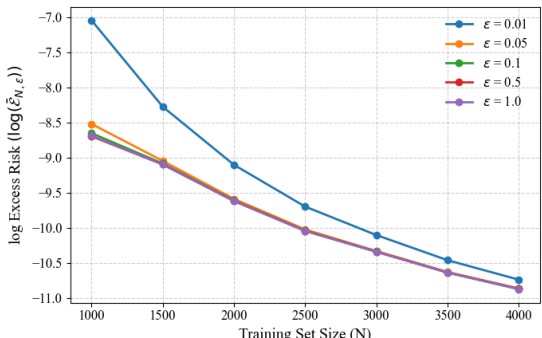

Figure 1: Excess risk of `NoisyHead` for the low-dimensional set-up as a function of training set size $N$ for different values of the privacy parameter $\varepsilon$ with $D = 5$. Two standard deviation error bars are also provided for each curve.

Figure 2: log Excess risk of `NoisyHead` for the high-dimensional set-up as a function of training set size $N$ for different values of the privacy parameter $\varepsilon$ with $D = \lfloor \sqrt{N} \rfloor$. Two standard deviation error bars are also provided after log transformation for each curve.

according to Theorem 4.1, with $\kappa = 1$ and $\delta = 10^{-5}$. The step size is set as $\eta_0 = 3.17/(5 + G^2)^2$, where $G$ denotes an upper bound on the norm of the projected features $\widetilde{Z}$, the ridge regularization parameter is fixed at $\lambda = 5$, and `NoisyHead` is trained for $T = \log N^{5/2}/\log(1 - \lambda\eta_0)$ iterations. We work in a noiseless setting with $\tau^2 = 0$.

For each $N$, the excess test risk for $T$ many iterations, averaged over $B = 500$ Monte-Carlo trials, is measured relative to that of ridge regression as

$$\hat{\mathcal{E}}_{N,\varepsilon} = \frac{1}{n_{\text{test}}} \sum_{k=1}^{n_{\text{test}}} \left( \langle \hat{\Gamma}_N - \Gamma^\star, Z_{k,\text{test}} \rangle \right)^2,$$

where $\hat{\Gamma}_N$ is the `NoisyHead` estimate based on $N$ training prompts, and $\Gamma^\star$ is the ridge regression solution. As shown in Figure 1, the excess risk decreases with $N$ and increases under stricter privacy, aligning with Theorem 4.2.

**High-dimensional regime.** We also consider the high-dimensional regime where $D \asymp L \asymp \sqrt{N}$. We vary $N \in \{500, 600, 700, 800, 900, 1000\}$ and $\varepsilon \in \{0.01, 0.05, 0.1, 0.5, 1.0\}$. The ridge regularization parameter is set as $\lambda = N/D$, and we use a fixed number of iterations $T = 5$ with step size $\eta = 0.07ND/(N + DG^2)^2$. All other parameters mirror those used in the low-dimensional setting. The average excess risk, computed over $B = 500$ repetitions, is reported in the right panel of Figure 2. The excess risk decreases with both $N$ and $\varepsilon$, though at a slower rate than in the low-dimensional case, consistent with Theorem 4.2 and reflecting the increased challenge of private learning in high dimensions.

### 6.2 Effect of early stopping in over-parametrized setting

In this section, we investigate how the number of gradient descent steps $T$ affects the test performance of the linear attention head trained using `NoisyHead`. In particular, we noted in Theorem 3.2 that as long as $T = \Theta(\log N)$, the "cost of descent", $C_{\text{Descent}} := \frac{D^2}{NL^2r}$ (ignoring log terms) in (4.5) dominates, which implies that initially, as $T$ increases, the right hand-side of (4.5) decreases. However, as $T$ is increased further, then the second term "cost of noise injection", $C_{\text{Noise}} := \frac{D\sqrt{L}}{N\log(NL)} \log^3 r$ starts to dominate, and the overall error of the Algorithm 1 starts to increase with $T$. This highlights the importance of stopping at an appropriate $T$, called "early stopping", for differentially private algorithm. Note that, among the two terms in (4.5), only the term $C_{\text{Noise}}$ owes itself to the differentially private updates; if $\sigma = 0$ in (4.2), we will always end up with the cost of descent as the net error of Algorithm 1.

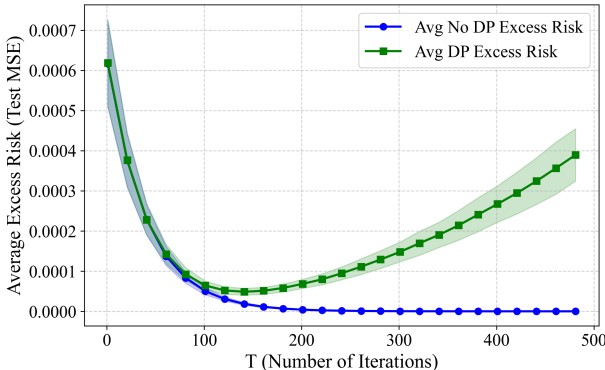

Figure 3: Interplay between the cost of descent and the cost of privacy in the overparameterized setting with $N = 1000$ and $\varepsilon = 0.8$.

As in Section 6.1, we work in a noiseless setting $\tau^2 = 0$. To compute $\mathrm{C}_{\mathrm{Descent}}$ and $\mathrm{C}_{\mathrm{Noise}}$, we adopt $N = 1000$, $L = \lfloor \sqrt{N} \rfloor$, $D = \lfloor \sqrt{N} \rfloor$, and vary $T \in \{1, 20, 40, \ldots, 480\}$. We further fix the regularization parameter at $\lambda = N/D$, correspondingly, we choose a step size of $\eta_0 = \frac{0.02\lambda}{3(\lambda + G^2)^2}$. For the DP parameters, we choose $\varepsilon = 0.8$ and $\delta = 10^{-5}$. For each value of $T$, we perform $B = 500$ Monte Carlo simulations as follows. For each $b \in [B]$, we generate $\mathcal{E}^{(b)} = (E_1^{(b)}, \ldots, E_N^{(b)})$, and generate $\Gamma_{\mathrm{DP}}^{T,(b)}$ by running the Algorithm 1. Moreover, we also run simple gradient descent (without the differential privacy setting) for $T$ many iterations to obtain $\Gamma_{\mathrm{Not\ DP}}^{T,(b)}$ to obtain a parameter estimate which is not differentially private. Let $\Gamma^\star$ be the corresponding ridge estimator. Note that, for a random test prompt $E$ and the corresponding $Z$, $(\langle \Gamma_{\mathrm{Not\ DP}}^{T,(b)} - \Gamma^\star, Z \rangle)^2$ is a proxy for $\mathrm{C}_{\mathrm{Descent}}$, the cost of descent. Finally, for each $b \in [B]$, we generate $n_{\mathrm{Test}} = 500$ many test prompts $E^{\mathrm{Test},(b)} := \{E_1^{\mathrm{Test},(b)}, \ldots, E_{n_{\mathrm{Test}}}^{\mathrm{Test},(b)}\}$, and compute

$$\hat{\mathrm{C}}_{\mathrm{Descent}}^{(b)} := \frac{1}{n_{\mathrm{Test}}} \sum_{l=1}^{n_{\mathrm{Test}}} (\langle \Gamma_{\mathrm{Not\ DP}}^{T,(b)} - \Gamma^\star, Z^{\mathrm{Test},(b)} \rangle)^2,$$

$$\hat{\mathrm{C}}_{\mathrm{Priv}}^{(b)} := \frac{1}{n_{\mathrm{Test}}} \sum_{l=1}^{n_{\mathrm{Test}}} (\langle \Gamma_{\mathrm{DP}}^{T,(b)} - \Gamma^\star, Z^{\mathrm{Test},(b)} \rangle)^2.$$

Note that the cost of privacy, $\hat{\mathrm{C}}_{\mathrm{Priv}}^{(b)}$ is an accumulation of both the cost of descent, $\hat{\mathrm{C}}_{\mathrm{Descent}}^{(b)}$, and the corresponding cost of noise injection. Finally, for each iteration $T$, we compute the total cost of privacy $\hat{\mathrm{C}}_{\mathrm{Priv}} := B^{-1} \sum_{b=1}^{B} \hat{\mathrm{C}}_{\mathrm{Priv}}^{(b)}$, and the cost of descent $\hat{\mathrm{C}}_{\mathrm{Descent}} := B^{-1} \sum_{b=1}^{B} \hat{\mathrm{C}}_{\mathrm{Descent}}^{(b)}$, and plot it against $T$. Figure 3 plots the evolution of two components of the prediction error: the *cost of descent* (blue) incurred by underoptimization, and the *cost of privacy* (green) due to noise injection. For small $T$, the descent cost dominates and the error decreases with additional optimization. However, beyond a critical number of iterations, the cost of privacy dominates, causing error to increase as more noise accumulates. This trade-off, predicted theoretically in Remark 4.2, yields a phase transition in the test error under privacy constraints. In contrast, in the non-private setting (approximating ridge regression), the error decreases monotonically with $T$.

## 6.3 Robustness of `NoisyHead`

Consider the setting of Section 5 with $\mu = 1$, and $\alpha = cN^p$ for $c \in \{2, 4\}$ and $p \in \{2, 2.02, 2.04, 2.06, 2.08, 2.1\}$, where we fix $N = 5000$, $L = 500$, $D = 5$, $\varepsilon = 0.5$, and $\delta = 10^{-2}$. We compare the ridge estimator $\Gamma^\star$ with the output of `NoisyHead` after $T = \log N$ iterations, using $\lambda = 0.01$ and step size $\eta_0 = 0.007/(0.01 + G^2)^2$, with all other parameters unchanged. Generalization error is measured via (5.3). In particular, we compute both sides of the errors in (5.3) for each pair of $(c, p)$ via $B = 500$ Monte Carlo simulations. For each $b \in [B]$, we simulate $\mathcal{E}_{\mathrm{Good}}^{(b)} := (E_{\mathrm{Good},1}^{(b)}, \ldots, E_{\mathrm{Good},N}^{(b)})$, and one bad prompt $E_{\mathrm{Bad},i}^{(b)}$ as in (5.1), upon selecting an $i_b \in [N]$

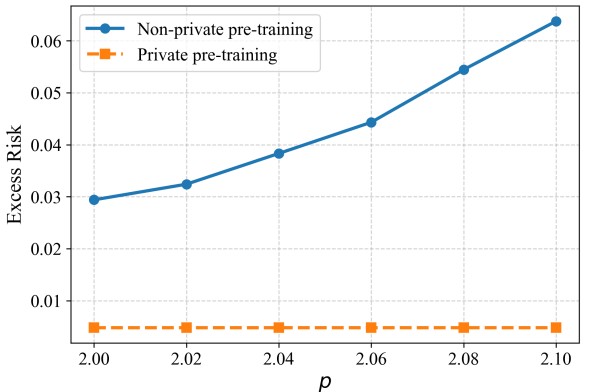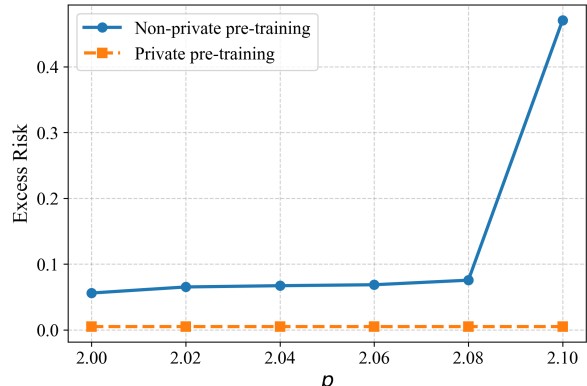

Figure 4: Comparison of prediction error under adversarial perturbations for different values of $c$. Left: $c = 2$; Right: $c = 4$. The differentially private estimator (`NoisyHead`) consistently outperforms the ridge estimator ($\Gamma^{\star}$) as the perturbation magnitude $\alpha = cN^p$ increases.

randomly. Let the corresponding perturbed set of prompts be denoted as

$$\mathcal{E}_{\text{bad}}^{(b)} := (E_{\text{Good},1}^{(b)}, \ldots, E_{\text{Good},i-1}^{(b)}, E_{\text{Bad},i}^{(b)}, E_{\text{Good},i+1}^{(b)}, E_{\text{Good},N}^{(b)}), \ b \in [B].$$

For each $b \in [B]$, we run Algorithm 1 and the ridge regression (2.7) on both $\mathcal{E}_{\text{Good}}^{(b)}$ and $\mathcal{E}_{\text{bad}}^{(b)}$ to obtain $\Gamma^{T,(b)}$, $\Gamma_{bad}^{T,(b)}$, $\Gamma^{\star,(b)}$ and $\Gamma_{bad}^{\star,(b)}$. Finally, to evaluate the performances of these models, we simulate $n_{\text{Test}} = 500$ many test prompts $E^{\text{Test},(b)}$ for each $b \in [B]$, and empirically estimate the two sides of (5.3) as follows

$$\text{Risk}_{DP} := B^{-1} \sum_{b=1}^{B} (\langle \Gamma^{T,(b)} - \Gamma_{bad}^{T,(b)}, Z^{\text{Test},(b)} \rangle)^2, \ \text{Risk}_{Ridge} := B^{-1} \sum_{b=1}^{B} (\langle \Gamma^{\star,(b)} - \Gamma_{bad}^{\star,(b)}, Z^{\text{Test},(b)} \rangle)^2.$$

Finally, we perform the experiment for each pair of $(c, p)$, and plot both the risks $\text{Risk}_{DP}$ and $\text{Risk}_{Ridge}$ for different values of $p$ and $c$. Figure 4 reports that, as $p$ increases, ridge regression becomes increasingly sensitive to the perturbation, while differentially private pretraining with `NoisyHead` remains substantially more robust. This validates Theorem 5.1 and its accompanying discussions in Remark 5.1.

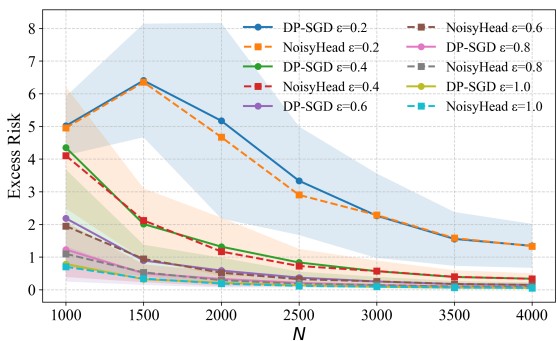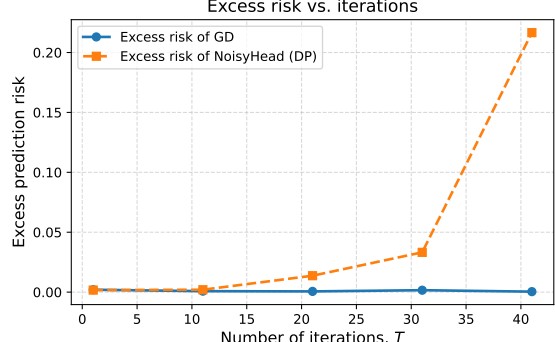

Figure 5: Excess risk of `NoisyHead` and DP-SGD for the low-dimensional set-up in Section 6.1 as a function of training set size $N$ for different values of the privacy parameter $\varepsilon$.

Figure 6: Excess risk of gradient descent on $\mathcal{L}_\lambda(\Gamma)$ and `NoisyHead` on the 20 Newsgroup dataset

### 6.4 Comparison with DP-SGD implemented in Opacus

To benchmark `NoisyHead` against the standard DP-SGD iterative scheme (with batch size 256), we implement a private optimization of the objective $\mathcal{L}_\lambda(\Gamma)$ in (2.6) using the `Opacus` library. We adopt the low-dimensional setup from Section 6.1, maintaining the same data-generating process and hyperparameter configurations for both methods. To ensure a fair comparison, we use identical iteration counts $T$, learning rates $\eta_0$, and clipping constants.

In Figure 5, we plot the excess prediction risks of both the methods relative to the non-private ridge oracle as a function of the training set size $N \in \{1000, \dots, 4000\}$ and privacy budget $\varepsilon \in \{0.2, \dots, 1.0\}$. Our results demonstrate that `NoisyHead` consistently matches or marginally outperforms the DP-SGD baseline across all configurations. This suggests that while `NoisyHead` achieves empirical parity with the standard pipeline, it uniquely provides a rigorous, closed-form privacy-utility analysis tailored specifically to the geometry of in-context learning.

## 7 Evaluation on the *20 Newsgroups* dataset

We evaluate the effectiveness of `NoisyHead` in learning information from text data within a natural language processing context by a prediction experiment on the *20 Newsgroups* dataset (Mitchell, 1997). This dataset contains approximately 50,000 news articles spanning multiple topical categories. For each article, we represent the text by its top-$D$ TF–IDF features (Rajaraman and Ullman, 2011), where $x \in \mathbb{R}^D$ denotes the TF–IDF scores of the $D$ most informative words, ordered by their importance within the article. The response variable $y \in \mathbb{R}$ corresponds to the article length (measured by the number of tokens). In our setting, we fix $D = 30$, resulting in token pairs $(x_i, y_i)^\top \in \mathbb{R}^{D+1}$ of total sequence length $L = 30$. The dataset is split into $N_{\text{train}} = 822$ training prompts and $N_{\text{test}} = 205$ test prompts.

The primary objective of this experiment is to investigate how the number of training iterations ($T$) influences the predictive performance of `NoisyHead` under differential privacy constraints. Specifically, we compare two predictors for the response $y$:

- the private predictor $\langle \widehat{\Gamma}_{\text{DP},T}, Z^{\text{Test}} \rangle$, where $\widehat{\Gamma}_{\text{DP},T}$ is obtained using `NoisyHead`; and

- the non-private predictor $\langle \widehat{\Gamma}_{\text{GD},T}, Z^{\text{Test}} \rangle$, where $\widehat{\Gamma}_{\text{GD},T}$ is produced by solving the ridge regression problem (2.6) using standard gradient descent without clipping or injected noise.

We take $\Gamma^\star$ to be the optimal ridge-regression solution and evaluate the *excess prediction risk* on the test data through

$$\text{Excess risk of NoisyHead} = \frac{1}{n_{\text{test}}} \sum_{k=1}^{n_{\text{test}}} \left( \langle \widehat{\Gamma}_{\text{DP},T} - \Gamma^\star, Z_k^{\text{Test}} \rangle \right)^2, \tag{7.1}$$

$$\text{Excess risk of GD} = \frac{1}{n_{\text{test}}} \sum_{k=1}^{n_{\text{test}}} \left( \langle \widehat{\Gamma}_{\text{GD},T} - \Gamma^\star, Z_k^{\text{Test}} \rangle \right)^2, \tag{7.2}$$

computed for ten equispaced values of $T$ between 1 and 100. The empirical results are summarized in Figure 6.

As illustrated in Figure 3, the test excess risk initially decreases or remains stable as the number of training iterations increases, reflecting improved fit during the early stages of learning. However, beyond a certain iteration threshold, the private model's performance deteriorates, while the non-private gradient-descent estimator continues to stabilize. This degradation in the private setting stems from the *accumulation of injected noise* across iterations, underscoring the importance of *early stopping* in differentially private training.

These empirical findings are in strong agreement with our theoretical results presented in Theorems 4.1 and 4.2, and further contextualized in Remark 4.1. Together, they demonstrate that the *privacy–utility trade-off* predicted by our analysis also extends to realistic NLP scenarios.

## 8 Conclusion

Maintaining privacy during the pretraining of attention-based models is an increasingly important challenge as such architectures become ubiquitous. To the best of our knowledge, this work provides the first systematic theoretical characterization of the privacy-utility trade-off in differentially private in-context learning using attention-based architectures. We quantify the impact of private pre-training on the in-context learning performance of linear attention heads and formally justify the necessity of *early stopping* (Zhang et al., 2023; Majmudar et al., 2022; Bu et al., 2024; Bombari and Mondelli, 2025) in the context of training attention-based models under differential privacy. A recent line of work has analyzed in-context learning performance of in architectures involving soft-max attention (Huang et al., 2023; Yang et al., 2024; Chen et al., 2024; Li et al., 2024a). It is an interesting future direction to extend our methods to such architectures and characterize the privacy-utility trade-off. Furthermore (Dai et al., 2023; Vladymyrov et al., 2024; Liang et al., 2025) show that multi-layered transformers can emulate gradient-based learning. Our framework offers a pathway toward understanding the theoretical behavior of such models when executing privacy-preserving pretraining, with potential implications for designing efficient pre-training mechanisms for complicated architectures to maintain the quality of responses as well as mitigate the "regurgitation" of private information about training subjects Carlini et al. (2021) often observed in large language models. Finally, an interesting direction is to explore the cost of privacy in next-token-prediction mechanisms adopted in language models utilizing attention-based transformer architectures.

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

# A  Theoretical Details

## A.1  Choice of parameters

[Yu et al. (2023)](#) provides an intuitive explanation for the behavior of the predictor (2.4). The term $w_{22}^{PV}W_{11}^{KQ}$ is approximately equal to $\mathbb{E}[(X^\top X)^{-1}]$, capturing the inverse second-moment structure of the features. The second term does not depend on the features, and the third term is independent of the labels $y$. They act as an extra additive term, which can be assumed to have no significant impact on the final prediction. The fourth term represents the effect of projecting the input features $x_i$ onto the direction $w_{12}^{PV}$ in the final prediction. However, since the features are assumed to be isotropic, it is reasonable to expect that projections onto any particular direction carry no special predictive value. Consequently, it is justified to assume that $w_{12}^{KQ} = 0$ and $w_{12}^{PV} = 0$, which simplifies the predictor to:

$$\hat{y}_{L+1} = \frac{1}{L}\left\langle x_{L+1}, w_{22}^{PV}W_{11}^{KQ}\sum_{i=1}^{L}y_i x_i \right\rangle.$$

This assumption is further supported by the observation in [Zhang et al. (2024)](#), where the authors show that when the parameters of $W^{PV}$ and $W^{KQ}$ are learned via gradient flow on the average reconstruction loss $\mathbb{E}[(\hat{y} - y)^2]$, initializing with $w_{12}^{KQ} = 0$ and $w_{12}^{PV} = 0$ ensures that the parameter remains zero throughout training.

## A.2  Proof of Theorem 3.2

Consider two datasets of prompts $(E_k)_{k\in[N]}$ and $(E_k')_{k\in[N]}$ that differ in exactly one prompt. Without loss of generality, assume $E_1 \neq E_1'$ and $E_k = E_k'$ for all $k \geq 2$. The change in the gradient update due to this single difference is bounded by:

$$\frac{\eta_0}{N}\left(\|\langle\Gamma^t, \widetilde{Z}_1\rangle\widetilde{Z}_1\|_F + \|\langle\Gamma^t, \widetilde{Z}_1'\rangle\widetilde{Z}_1'\|_F + \|\texttt{clip}_{\mathcal{C}}(y_{1,L+1})\widetilde{Z}_1\|_F + \|\texttt{clip}_{\mathcal{C}}(y_{1,L+1}')\widetilde{Z}_1'\|_F\right)$$
$$\leq \frac{2\eta_0(RG^2 + \mathcal{C}G)}{N} \leq \frac{\eta_0\sigma}{N}, \tag{A.1}$$

where the final inequality follows from the assumption on $\sigma$.

By Lemma 2.5 of [Kamath and Ullman (2020)](#), each gradient step in Algorithm 1 is $(\varepsilon/T, \delta/T)$-differentially private. The overall guarantee then follows by composition, using Fact 2.2 of [Cai et al. (2021)](#).

## A.3  Proof of Theorem 4.1

Consider the set $\mathcal{D}_1 := \{\|\Gamma^\star\|_F \leq R\}$. Moreover, denote

$$\widetilde{\Gamma}^{t+1} = (1 - 2\lambda)\Gamma^t - \eta_0 N^{-1}\sum_{k=1}^{N}\left(\langle\Gamma^t, \widetilde{Z}_k\rangle - \texttt{clip}(y_{L+1})\right)\widetilde{Z}_k.$$

Clearly, $\Gamma^{t+1} = \Pi_R(\widetilde{\Gamma}^{t+1} + \boldsymbol{z}_t)$. Under $\mathcal{D}_1$, it is easy to see that

$$\|\hat{\Gamma} - \Gamma^\star\|_F^2 \leq \|\widetilde{\Gamma}^T + \boldsymbol{z}_{T-1} - \Gamma^\star\|_F^2 \leq (1 + C_0^{-1})\|\widetilde{\Gamma}^T - \Gamma^\star\|_F^2 + (1 + C_0)\|\boldsymbol{z}_{T-1}\|_F^2, \tag{A.2}$$

where, the choice of the constant $C_0$ ensures

$$(1 + C_0^{-1})\kappa < 1 - \eta_0\lambda\,, \text{ with } \kappa := 1 - 2\eta_0\lambda + \eta_0^2(G^2 + 2\lambda)^2. \tag{A.3}$$

Further consider the sets $\mathcal{D}_2 := \left\{\max_{k\in[N]}\left\|\sum_{i=1}^{L}x_{k,i}\right\|_2 \leq GL\mathcal{C}^{-1}\right\}$, and $\mathcal{D}_3 := \left\{\max_{k\in[N],i\in[L+1]}|y_{k,i}| \leq \mathcal{C}\right\}$. Since $\|x_{k,L+1}\|_2 = 1$, under the events $\mathcal{D}_2$ and $\mathcal{D}_3$, it follows that

$$\max_{k\in[N]}\left\|L^{-1}x_{k,L+1}\sum_{i=1}^{L}y_{k,i}x_{k,i}^\top\right\|_F \leq G, \tag{A.4}$$

which implies $\widetilde{Z}_k = Z_k$ for all $k \in [N]$ by the definition of $Z_k$ in (2.5). The sets $\mathcal{D}_i, i = 1, 2, 3$ allow us to bear down the classical theory of convex minimization, and our choice of the parameters $R, G$ and $\mathcal{C}$ will emphasize that these events occur with high probability. In particular, under $\mathcal{D} := \cap_{i=1}^3 \mathcal{D}_i$, we note the $\mathcal{L}$ is $\lambda$-strongly convex:

$$\langle \nabla_\Gamma \mathcal{L}(\Gamma, (Z_k)_{k \in [N]}), \Gamma - \Gamma^\star \rangle \geq \lambda \|\Gamma - \Gamma^\star\|_F^2, \tag{A.5}$$

and the $(G^2 + 2\lambda)$-smooth:

$$\left\| \nabla_\Gamma \mathcal{L}(\Gamma, (Z_k)_{k \in [N]}) - \nabla_\Gamma \mathcal{L}(\Gamma', (Z_k)_{k \in [N]}) \right\|_F \leq (G^2 + 2\lambda) \left\| \Gamma - \Gamma' \right\|_F. \tag{A.6}$$

Therefore, for the term $\|\widetilde{\Gamma}^T - \Gamma^\star\|_F$ in (A.2),

$$\|\widetilde{\Gamma}^T - \Gamma^\star\|_F^2 = \|\Gamma^{T-1} - \eta_0 \nabla_{\Gamma^{T-1}} \mathcal{L}(\Gamma^{T-1}, (Z_k)_{k \in [N]}) - \Gamma^\star\|_F^2 \leq \kappa \|\Gamma^{T-1} - \Gamma^\star\|_F^2, \tag{A.7}$$

where we recall $\kappa$ from (A.3), and (A.7) employs (A.5) and (A.6). Note that we must require $\kappa < 1$, which makes use of $\eta_0 < \frac{\lambda}{(G^2 + 2\lambda)^2}$. Putting (A.7) back into (A.2), one obtains under $\mathcal{D}$ that

$$\|\hat{\Gamma} - \Gamma^\star\|_F^2 \leq (1 + C_0^{-1})\kappa \|\Gamma^{T-1} - \Gamma^\star\|_F^2 + (1 + C_0)\|\boldsymbol{z}_{T-1}\|_F^2.$$

Proceeding recursively, we can show that for all $T > 1$, we have

$$\|\hat{\Gamma} - \Gamma^\star\|_F^2 \leq (1 - \eta_0\lambda)^T R^2 + (1 + C_0) \sum_{i=0}^{T-1} (1 - \eta_0\lambda)^{T-i-1} \|\boldsymbol{z}_i\|_F^2. \tag{A.8}$$

Since the errors $(\boldsymbol{z}_i)_{i=1}^T$ are independent of the prompts $(E_k)_{k \in [B]}$, an application of Lemma A.2. of Cai et al. (2021) implies

$$\|\hat{\Gamma} - \Gamma^\star\|_F^2 \lesssim (1 - \eta_0\lambda)^T R^2 + \sigma^2 \eta_0^2 D \frac{T^2 \log(2T/\delta)}{N^2 \varepsilon^2},$$

with probability at least $1 - c_1 \exp(-c_2 D)$ under $\mathcal{D}$. An application of Cauchy-Schwarz inequality entails

$$(\langle \hat{\Gamma}, Z \rangle - \langle \Gamma^\star, Z \rangle)^2 \leq G_0^2 \left( (1 - \eta_0\lambda)^T R^2 + \sigma^2 \eta_0^2 D \frac{T^2 \log(2T/\delta)}{N^2 \varepsilon^2} \right) \tag{A.9}$$

with probability at least $1 - c_1 \exp(-c_2 D)$ under $\mathcal{D} \cap \{\|Z\|_F \leq G_0\}$. Now we turn to tackling the individual events $\mathcal{D}_i, i = 1, 2, 3$. For $\mathcal{D}_3$, note that if $(x_i)_{i \in [L]} \overset{i.i.d.}{\sim} \mathcal{U}(\mathbb{S}^{D-1})$ and $w \sim N(0, \mathbb{I}_D)$ independently of $x_i$'s, then $(w^\top x_i) \sim N(0, 1)$ marginally. Therefore, Lemma A.1 implies

$$\mathbb{P}(\mathcal{D}_3) \geq 1 - \kappa, \quad \text{for } \mathcal{C} = \sqrt{2\nu \log(NL/\kappa)}. \tag{A.10}$$

Furthermore, with $G \asymp \frac{\mathcal{C}}{\sqrt{L}} \left( 1 + \left( \frac{\log(N/\kappa)}{D^2} \right)^{1/4} \right)$, from Lemma A.2 we get that

$$\mathbb{P}(\mathcal{D}_2) \geq 1 - \kappa. \tag{A.11}$$

Finally, noting that

$$\sum_{k=1}^N y_{k,L+1} \text{vec}(Z_k) \leq \max_{k \in [N]} |y_{k,L+1}| \left( \max_{k \in [N], i \in [L]} |y_{k,i}| \right) \frac{1}{L} \sum_{k,i} \text{vec}(x_{k,L+1} x_{k,i}^\top),$$

an application of Lemma A.1 on (2.7), in conjunction with Lemma A.3, yields

$$\mathbb{P}(\mathcal{D}_1) \geq 1 - \kappa, \text{ with } R = \lambda^{-1} \mathcal{C}^2 \sqrt{\frac{N}{L}} \left( 1 + \left( \frac{\log(1/\kappa)}{D^2} \right)^{1/4} \right). \tag{A.12}$$

Finally, similar to Lemma A.2 it can be argued that

$$\mathbb{P}(\|Z\|_F \le G_0) \ge 1 - \kappa, \quad \text{for } G_0 \asymp \frac{\mathcal{C}}{\sqrt{L}} \left(1 + \left(\frac{\log(1/\kappa)}{D^2}\right)^{1/4}\right). \tag{A.13}$$

Summarizing (A.10)-(A.12), it holds that

$$\mathbb{P}(\mathcal{D} \cap \{\|Z\|_F \le G_0\}) \ge 1 - 4\kappa. \tag{A.14}$$

Putting these bounds back into (A.9), we invoke (A.14) to conclude (4.2).

## A.4 Proof of Theorem 4.2

(i) *Low-dimensional setting.* Recall $T = \frac{\log(N^2 L D^3)}{\log((1-\eta_0\lambda)^{-1})}$. Note that, with $\kappa > e^{-D^2}$, we have $G_0 \lesssim \mathcal{C}/\sqrt{L}$, $R \lesssim \lambda^{-1}\mathcal{C}^2\sqrt{N/L}$, $\eta_0 \asymp \frac{\lambda}{(\lambda+G^2)^2} \lesssim 1/\lambda$ and $\lambda \asymp 1 \asymp \eta_0$ from (4.1). Hence, the first term of (4.3) can be bounded as

$$G_0^2(1-\eta_0\lambda)^T R^2 \lesssim \frac{\mathcal{C}^6}{NL^3D^3} \lesssim \nu^3 \log^3 \frac{NL}{\kappa} \frac{1}{NL^3D^3}.$$

Moreover, from $\log(\frac{1}{\kappa}) \lesssim D^2 \lesssim \log(NL)$, we have that

$$G_0^2(1-\eta_0\lambda)^T R^2 \lesssim \nu^3(\log^3 NL) \cdot \frac{1}{NL^3}. \tag{A.15}$$

On the other hand, write the second term as

$$G_0^2\sigma^2\eta_0^2 D \frac{T^2}{N^2} \frac{\log(2T/\delta)}{\varepsilon^2} \lesssim \frac{\mathcal{C}^2}{L}(\mathcal{C}G + RG^2)^2 D \frac{T^3}{N^2} \frac{\log(1/\delta)}{\varepsilon^2}. \tag{A.16}$$

Clearly, for $\sigma$, one obtains,

$$\mathcal{C}G + RG^2 \lesssim \frac{\mathcal{C}^2}{\sqrt{L}}\left(1 + \left(\frac{\log(NL)}{D^2}\right)^{1/2}\right) + \mathcal{C}^2\sqrt{\frac{N}{L}}\frac{\mathcal{C}^2}{L}\left(1 + \left(\frac{\log(NL)}{D^2}\right)\right)$$

$$\lesssim \nu^2 \log^3(NL)\frac{\sqrt{N}}{LD^2},$$

where the second inequality is attained by using $\log(N/\kappa) = \log N + \log 1/\kappa \lesssim \log N + D^2 \lesssim \log NL$, and the final assertion follows from $(\log NL)/D^2 >> (\sqrt{\log NL})/D$. Therefore, from (A.16), the second term is bounded by,

$$\lesssim \frac{\nu^5 \log^7(NL)}{L} \cdot \frac{N}{L^2D^4} \cdot D \cdot \frac{\log^3(N^2LD^3)}{N^2} \cdot \frac{\log(1/\delta)}{\varepsilon^2}$$

$$\lesssim \frac{\nu^5 \log^7(NL)\log^3(N^2LD^3)}{NL^3D^3} \cdot \frac{\log(1/\delta)}{\varepsilon^2}$$

$$\lesssim \frac{\nu^5 \log^{10}(NL)}{NL^3} \cdot \frac{\log(1/\delta)}{\varepsilon^2}. \tag{A.17}$$

Combining (A.15) and (A.17) yields the proof for the low-dimensional case.

(ii) *High-dimensional setting.* Here, $\kappa \gtrsim (NL)^{-1}$, and $D^2 \gtrsim \log(NL)$ implies that $\frac{\log(NL/\kappa)}{D^2} \lesssim 1$. We also have $G \lesssim \mathcal{C}/\sqrt{L}$, $G_0 \lesssim \mathcal{C}/\sqrt{L}$ and, $R \lesssim \lambda^{-1}\mathcal{C}^2\sqrt{N/L}$. The first term of (4.4) can be bounded as

$$G_0^2 R^2(1-\eta_0\lambda)^T \lesssim \frac{\mathcal{C}^2}{L} \cdot \mathcal{C}^4 \frac{N}{L} \cdot \frac{1}{\lambda^2} \cdot \frac{1}{r} \lesssim \nu^3 \log^3(NL) \cdot \frac{N}{L^2\lambda^2 r}. \tag{A.18}$$

Furthermore, for the second term, observe that $\eta_0 \lesssim 1/\lambda$, and

$$\sigma = G(\mathcal{C} + RG) \asymp \frac{\mathcal{C}^2}{\sqrt{L}} + \mathcal{C}^2 \sqrt{\frac{N}{L}} \cdot \frac{1}{\lambda} \cdot \frac{\mathcal{C}^2}{L} = \frac{\mathcal{C}^2}{\sqrt{L}} \left(1 + \frac{\sqrt{N}}{\lambda L} \mathcal{C}^2\right).$$

Therefore, the second term can be bounded as

$$\begin{aligned}
G_0^2 \sigma^2 \eta_0^2 D \frac{T^3}{N^2} &\leq \frac{\mathcal{C}^2}{L} \cdot \sigma^2 \frac{1}{\lambda} D \cdot \frac{T^3}{N^2} \leq \frac{\mathcal{C}^2}{L} \cdot \log^3 r \cdot \frac{D}{N^2 \lambda^2} \cdot \sigma^2 \\
&\leq \frac{\mathcal{C}^2}{L} \log^3 r \frac{D}{N^2 \lambda^2} \cdot \frac{\mathcal{C}^4}{L} \left(1 + \frac{N}{\lambda^2 L^2} \mathcal{C}^4\right) \\
&\leq \frac{\mathcal{C}^6 D}{N^2 L^2 \lambda^2} \left(1 + \frac{N}{\lambda^2 L^2} \log^2(NL)\right) \log^3 r \\
&\lesssim \nu^3 \frac{N \log^3 r}{L^2 \lambda^2} \log^3 NL \frac{D}{N^3} \left(1 + \frac{N}{\lambda^2 L^2} \log^2 NL\right). \qquad (A.19)
\end{aligned}$$

Assertions (A.18) and (A.19) conclude the proof.

## A.5 Proof of Proposition 4.1

From $\lambda \asymp N/D \asymp \sqrt{N}$, it follows that $\frac{D^2}{NL^2} \asymp \frac{N}{L^2 \lambda^2} \asymp N^{-1}$, and hence, $\frac{\log^2(NL)}{\lambda^2} \lesssim 1$. Therefore, from (4.4), (4.5) follows trivially. Further, the first term in (4.5) dominates as long as $r \log^3 r \ll \frac{N^3}{D}$, yielding (4.6) when $r \asymp N$.

## A.6 Proof of Theorem 5.1

Recall the definition of $G_0$ and $R$ from Theorem 4.1. In view of $\kappa > N \exp(-D^2)$, (A.4) and $\|\hat{\Gamma}\|_F \vee \|\Gamma^{T^{\mathrm{bad}}}\|_F \leq R$, using (A.13), it holds that

$$(\langle \hat{\Gamma}, Z \rangle - \langle \hat{\Gamma}_{\mathrm{bad}}, Z \rangle)^2 \leq \frac{\mathcal{C}^2}{L} R^2, \qquad (A.20)$$

with probability at least $1 - \kappa$. On the other hand, for the analysis of the ridge estimates, recall (2.7). Clearly, from Lemma A.3, it holds with probability $\geq 1 - 2\kappa$ that

$$\left\| \sum_{k=1}^{N} \mathrm{vec}(Z_k) \mathrm{vec}(Z_k)^\top \right\|_F \leq \mathcal{C}^2 \frac{N}{L} < N\lambda, \qquad (A.21)$$

where the final equality follows from $\lambda > \mathcal{C}^2 L^{-1}$. Moreover, from Lemma A.2, it holds with probability $\geq 1 - 2\kappa$ that

$$\left\| \mathrm{vec}(Z_{\mathrm{bad},i}) \mathrm{vec}(Z_{\mathrm{bad},i})^\top \right\|_F \lesssim \frac{\mathcal{C}^2}{L} + \frac{\alpha^2 \mu^4}{L} \asymp \frac{\alpha^2 \mu^4}{L} \lesssim N\lambda, \qquad (A.22)$$

where the first part of the inequality follows from the lower bound on $\alpha^2 \mu^2$ and the second inequality follows from the upper bound on $\alpha^2 \mu^4$ as stated in (5.2). Consequently, combining (2.7) with (A.21) and (A.22) jointly yields,

$$\| \mathrm{vec}(\Gamma_{\mathrm{bad}}^\star - \Gamma^\star) \| \geq \frac{\|y'_{i,L+1} \mathrm{Vec}(Z'_i) - y_{i,L+1} \mathrm{Vec}(Z_i)\|}{N\lambda} \qquad (A.23)$$

with probability at least $1 - 4\kappa$. Since $\|y_{i,L+1} \mathrm{vec}(Z_i)\| \leq \frac{\mathcal{C}^2}{\sqrt{L}}$ with probability at least $1 - \kappa$, invoking (5.2), yet another application of Lemma A.2 yields

$$\|y'_{i,L+1} \mathrm{Vec}(Z_{\mathrm{bad},i}) - y_{i,L+1} \mathrm{Vec}(Z_i)\| \geq \alpha^2 \mu^2 \qquad (A.24)$$

with probability at least $1 - \kappa$. Since (5.2) also implies $\frac{\alpha^2 \mu^2}{N\lambda} > R^2 \frac{\mathcal{C}^2}{L}$, from (A.20), (A.22), (A.23), and (A.24), we obtain (5.3).

### A.7 Auxiliary Lemmas

The following lemmas are instrumental to proving our theorems 4.1 and 5.1, and hereby are listed. In particular, Lemma A.1 and A.3 follows using Hoeffding's inequality and a union bound argument.

**Lemma A.1.** *If $z_{kj} \sim N(0, 1 + \tau^2)$ $k \in [N], j \in [L]$ are not necessarily independent, then*

$$\mathbb{P}\left( \max_{k,j} |z_{ij}| \lesssim \sqrt{(1 + \tau^2) \log\left(\frac{4NL}{\kappa}\right)} \right) \geq 1 - \kappa.$$

For the Lemmas A.2 and A.3, note that for any vector $x \in \mathbb{R}^D$, the Euclidean norm $\|x\|_2 = \sup_{a \in \mathbb{S}^{D-1}} a^\top x$. For any fixed $a \in \mathbb{S}^{D-1}$ and $k \in [N]$, $a^\top \sum_{i=1}^L x_{k,i}$ is a sub-Gaussian random variable with variance proxy $L/D$ . Therefore

$$\mathbb{P}\left[ \left| a^\top \sum_{i=1}^L x_{k,i} \right| > \sqrt{L/D}\, t \right] \lesssim \exp\left( -t^2 \right).$$

Therefore, using a covering number argument similar to Theorem 1.19 of Rigollet and Hütter (2023) one can show the following.

**Lemma A.2.** *Suppose $(x_{k,i})_{k \in [N], i \in [L]} \overset{i.i.d.}{\sim} \mathcal{U}(\mathbb{S}^{D-1})$. Then,*

$$\mathbb{P}\left( \max_{k \in [N]} \|\sum_{i=1}^L x_{k,i}\|_2 \lesssim \sqrt{L}(1 + D^{-1/2}(\log(\frac{N}{\kappa}))^{1/2}) \right) \geq 1 - \kappa.$$

**Lemma A.3.** *Suppose $(x_{k,i})_{k \in [N], i \in [L]} \overset{i.i.d.}{\sim} \mathcal{U}(\mathbb{S}^{D-1})$. Then,*

$$\mathbb{P}\left( \|\sum_{k=1}^N \sum_{i=1}^L \text{vec}(x_{k,L+1} x_{k,i}^\top)\|_2 \lesssim \sqrt{NL}(1 + D^{-1}(\log(\frac{1}{\kappa}))^{1/2}) \right) \geq 1 - \kappa.$$

## B Comparing `NoisyHead` with differentially private ridge regression

Since our loss function in training linear attention head takes the form of a ridge regression (see, (2.6)-(2.7)), an alternative approach to estimating $\Gamma$ with $(\varepsilon, \delta)$-DP guarantee can be computing the optimal solution to the optimization problem in (2.6) and adding Gaussian error with noise variance calibrated to the $\ell_2$ sensitivity of the estimator.

In this section, we provide a numerical study to compare the prediction risk of the ridge-regression based estimator and `NoisyHead`. By retracing steps in the proof of Theorem 4.1, it can be argued that the $\ell_2$ sensitivity of the ridge estimator is bounded above, with high probability, by $\frac{CG}{\lambda N}$, where $N$ is the number of training prompts, $\lambda$ is the regularization parameter, and $\mathcal{C}$ and $G$ are the clipping and projection parameters defined in Theorem 4.1. In light of this, consider the ridge estimator:

$$\hat{\Gamma}^\dagger = \left( \lambda N I + \sum_{k=1}^N \text{vec}(\tilde{Z}_k) \text{vec}(\tilde{Z}_k)^\top \right)^{-1} \sum_{k=1}^N \text{clip}_{\mathcal{C}}(y_{k,L+1}) \text{vec}(\tilde{Z}_k),$$

where $\tilde{Z}$ and $\text{clip}_{\mathcal{C}}$ are defined in `NoisyHead` in Algorithm 1. One can consider the following noisy estimator of $\Gamma$:

$$\hat{\Gamma}_{\text{ridge}}^\dagger := \hat{\Gamma}^\dagger + W^\dagger, \qquad \text{where } W^\dagger \sim \mathcal{N}_{D^2}\left( 0, \frac{\mathcal{C}^2 G^2 \log(1.25/\delta)}{\lambda^2 N^2 \varepsilon^2} \mathbb{I}_{D^2} \right).$$

For the *low-dimensional* setting as in Section 6.1, we provide empirical results on the prediction risk in unseen test prompts for the `NoisyHead`-based mechanism (Algorithm 1) versus predicting the response in the test prompt by $\langle Z_{\text{test}}, \hat{\Gamma}_{\text{ridge}}^\dagger \rangle$. The excess risk of prediction (compared non-DP baseline) for either method in the

configurations considered in Section 6.1 is summarized in Table 1. Evidently, the direct noise injection into the ridge estimate outperforms the `NoisyHead` estimate in terms of prediction accuracy. This phenomenon can be explained by the early stopping mechanism observed in the foregoing sections, which is essential to prevent excessive noise accumulation. This keeps the final estimate away from the true optima, resulting in additional bias in the estimates.

| $N$ | $\varepsilon$ | NoisyHead | DP-ridge |
|------|------|---------|-----------------------|
| 2000 | 0.2 | 0.1302  | $5.86 \times 10^{-5}$ |
| 2000 | 0.4 | 0.1305  | $1.45 \times 10^{-5}$ |
| 3000 | 0.2 | 0.07280 | $9.73 \times 10^{-6}$ |
| 3000 | 0.4 | 0.03517 | $2.44 \times 10^{-6}$ |
| 4000 | 0.2 | 0.02597 | $2.71 \times 10^{-6}$ |
| 4000 | 0.4 | 0.00652 | $6.75 \times 10^{-7}$ |

Table 1: Comparison of prediction error under `NoisyHead` and DP-ridge setting

Nevertheless, the ridge regression-based estimation procedure is critically tuned to the linear attention framework and does not provide any insights into the cost-of-privacy in the training of attention based mechanism on the performance in in-context learning. This limits extending such insights to the training of generic architectures such as those relying on soft-max attention. In contrast, `NoisyHead` follows the standard gradient descent-based approach in training predictive models. In particular, controlling the sensitivity of gradients is easier for a wide range of architectures, whereas controlling the sensitivity of the final estimate can be extremely hard in a more general setting. Furthermore, the scaling laws for the excess risk in either framework will differ only by logarithmic factors if $T$, $\lambda$ and $\eta_0$ are appropriately chosen.

