# OpenReview forum: "How Private is Your Attention? Bridging Privacy with In-Context Learning"
_TMLR — Accepted by TMLR_

### Review · Reviewer_jHSy · 2025-12-02

**Summary Of Contributions:**

This paper investigates how enforcing differential privacy (DP) during pretraining affects the in-context learning (ICL) performance of attention-based models in a stylized linear regression setting. The authors focus on a linear attention head that performs ICL over sequences of feature–label pairs, where the task is to predict the label of a query point given its features and a set of in-context examples (i.e., features and labels drawn from the same linear regression task). The paper proposes NoisyHead, a differentially private pretraining procedure for this attention head. Specifically, NoisyHead applies a modified gradient descent on a regularized squared loss over pretraining prompts, incorporating (i) clipping of responses and sufficient statistics, (ii) projection of parameters onto a ball, and (iii) Gaussian noise injected into each gradient step. The authors then analyze the cost of privacy, measured as the excess prediction risk of the DP estimator relative to the non-private ridge solution. In addition, they analyze the robustness of NoisyHead. Numerically, the paper presents simulations and a real-data experiment to assess the effectiveness of NoisyHead and to validate the theoretical results.

Key Strengths:

- The paper is overall well written and well structured.

- The question studied is interesting and timely. The work provides a DP algorithm and analysis tailored to in-context learning, rather than classical supervised learning, which may yield valuable insights for current LLM pretraining.

- The analysis is technically nontrivial and appears sound.

Key Weaknesses:

- My main concern is that the model is highly stylized: linear regression with features on the unit sphere, Gaussian task parameters and noise, and a single linear attention head. While I appreciate the clean analysis of the proposed algorithm, there remains a substantial gap to realistic transformers and data distributions for more complex tasks.

- The experiments are mostly synthetic and focused on validating asymptotic behavior, and the single real-data experiment is simple and does not directly involve language modeling or prompts in the usual sense. Moreover, there is no empirical comparison against other DP training schemes (e.g., naive DP-SGD or DP ridge regression at the task level), which would better position NoisyHead in the DP literature.

**Audience:**

Yes

**Audience Explanation:**

This work sits at the intersection of two active areas for the TMLR community:

- In-context learning and attention mechanisms. There is a line of work on understanding how transformers implement ICL and emulate algorithms such as regression and gradient descent. This paper extends this line of research by incorporating differential privacy into the picture.

- Differential privacy for large models. DP-SGD and related techniques are widely studied for deep networks, but there is relatively limited theory on how privacy constraints interact with ICL behavior.

Although the setup is simplified (linear regression with a linear attention head), many works in TMLR and top conferences use stylized models to gain insight into phenomena observed in large-scale systems. The results here provide conceptual guidance that could inform the design and analysis of DP training procedures for more realistic transformer architectures. For these reasons, I believe a nontrivial portion of the TMLR audience would find these findings relevant.

**Broader Impact Concerns:**

I do not see any such concerns.

**Claims And Evidence:**

Yes

**Claims Explanation:**

On the theoretical side, the main claims (e.g., privacy guarantees and cost-of-privacy bounds) are stated as formal theorems with detailed proofs. Empirically, the synthetic experiments are well aligned with the theory: the authors choose algorithm hyperparameters according to their theorems and demonstrate the decay of excess risk in both low- and high-dimensional regimes (Figures 1 and 2). The 20 Newsgroups experiment (Figure 5) is small but convincingly demonstrates the same early-stopping phenomenon in a non-synthetic setting.

**Requested Changes:**

- For the numerical experiments, since the introduction explicitly contrasts NoisyHead with DP-SGD (e.g., treating full sequences as private vs. summarizing prompts), it would be very valuable to include an empirical comparison to a straightforward DP-SGD baseline on the same linear head. At a minimum, the paper should provide a careful discussion explaining why such a baseline is technically challenging to implement or would trivially underperform, quantified as much as possible using existing DP-SGD analyses.

- I also think it would help a broader audience from the LLM community if the paper more clearly connects the current setup to realistic large language models. For example, what is the relevant privacy unit in practical applications (a user, a document, a session of interactions)? How does prompt-level DP relate to the kind of privacy protection we seek in language models? A more explicit discussion along these lines would make the implications of the results easier to interpret.

---

> ### Author Response · Authors · 2026-02-12
> **Follow up on the rebuttal**
>
> Dear reviewer,
>
> Since the deadline of February 15 is approaching, kindly let us know if you have any further comments, which we will be happy to address.
>
> Best regards
> Authors

---

> > ### Comment · Reviewer_jHSy · 2026-02-13
> >
> > Thank you for the revisions and for adding the additional empirical comparisons. However, my concern is not fully resolved. While I appreciate the clarity of the toy-model analysis, I was hoping for a more explicit discussion mapping the privacy unit to realistic units in practice (e.g., a user, document, or interaction session). If I have missed such a discussion, please point me to it. Generally, more explicit point-by-point responses to reviewer comments would be helpful.

---

> ### Author Response · Authors · 2026-02-14
> **Response to the reviewer's concern**
>
> >Thank you for the revisions and for adding the additional empirical comparisons. However, my concern is not fully resolved.
> While I appreciate the clarity of the toy-model analysis, I was hoping for a more explicit discussion mapping the privacy unit to realistic units in practice (e.g., a user, document, or interaction session). If I have missed such a discussion, please point me to it.
>
> We thank the reviewer for requesting a clearer practical interpretation of the privacy unit. In our framework, a prompt corresponds to a realistic user-level interaction session. For example, consider training a foundation model to predict cancer severity scores from patient history and medical imaging. Suppose $N$ hospitals participate, and each hospital provides a prompt consisting of $L$ labeled patient cases (feature embeddings derived from clinical history and imaging together with clinician-annotated severity scores), along with a query patient whose true score is used during pretraining. Differences in patient populations and diagnostic practices across hospitals induce different latent regression parameters, making this naturally an in-context learning problem rather than classical regression. Each prompt is sensitive in its entirety, since leakage of any component could reveal private patient information. Our $(\varepsilon,\delta)$-DP guarantee therefore protects at the level of a whole prompt, ensuring that the released attention head is nearly indistinguishable whether or not any single hospital’s complete interaction session is included in training. We are open to adding this explanation in the introduction at your suggestion. Please also see the response to Comment 1 of reviewer oayZ.

---

> > ### Author Response · Authors · 2026-02-14
> > **Response to reviewer's concern on point-by-point response**
> >
> > > Generally, more explicit point-by-point responses to reviewer comments would be helpful.
> >
> > Thanks for this suggestion. In our previous response, we have already provided a point-by-point response to the requested changes. Below we provide the same for key weaknesses.
> >
> > >  My main concern is that the model is highly stylized: linear regression with features on the unit sphere, Gaussian task parameters and noise, and a single linear attention head. While I appreciate the clean analysis of the proposed algorithm, there remains a substantial gap to realistic transformers and data distributions for more complex tasks.
> >
> > We have included a discussion on this in our rebuttal to point 1 of the reviewer ZfeQ. We reproduce it for your convenience.
> >
> > We thank the reviewer for this remark. While it is true that we analyze a simplified architecture that intends only to address private pre-training of a linear attention head to ICL in linear regression problems, this choice is intentional and motivated by the fact that, even in this simplified setting, the trade-off between differential privacy (DP) and accuracy has remained poorly understood. Prior to our work, there was no known characterization of the privacy–utility frontier in this regime.
> >
> > Our contribution addresses this fundamental gap by providing, to our knowledge, the first near-optimal bound capturing the interplay between privacy guarantees and predictive performance in in-context learning with linear transformers. This result not only establishes a baseline for privacy-preserving learning in a controlled setting, but also serves as a theoretical stepping stone for studying more complex architectures and tasks.
> >
> > In a similar spirit, foundational works in the non-private setting -such as those by Garg et al. [‘What Can Transformers Learn In-Context? A Case Study of Simple Function Classes’, NeurIPS, 2022], Zhang et al. ([‘Trained Transformers Learn Linear Models In-Context’, JMLR, 2024, ‘In-Context Learning of a Linear Transformer Block’, NeurIPS, 2024]) - began by analyzing linear transformers and regression tasks to understand the capabilities of in-context learning before extending to richer scenarios. We believe that building a rigorous understanding in this stylized regime (as exemplified in many works including the works mentioned above) is a crucial prerequisite for meaningful generalizations to deep, non-linear transformers trained under privacy constraints.
> >
> > We admit that our work may not address all the concerns of the empirical ML community, where more complex architectures are usually studied; in this paper, our goal is to provide a theoretical understanding of the cost of privacy in pre-training attention heads. Therefore, for mathematical convenience, we start with a simpler set-up to accurately and rigorously characterize the privacy-utility trade-off. As is common in theoretically underdeveloped areas, such as the emerging theory of transformers, our objective is not to attain state-of-the-art benchmark accuracy, but rather to provide principled theoretical insight into mechanisms that may govern more complex empirical behavior. Indeed, a rigorous mathematical characterization of prediction risk in in-context learning (ICL) for general transformer architectures remains far from universal, and is poorly understood even in the non-private setting. We view this work as an initial step toward a broader theoretical understanding of privacy enforcement in modern attention-based models; however, extending such guarantees to more realistic architectures (including soft-max attention) is likely to require substantially more intricate and context-dependent mathematical analysis. In the conclusion, we have included studying the privacy-utility trade-off in more generic transformer architectures as a potential future direction.
> >
> > > The experiments are mostly synthetic and focused on validating asymptotic behavior, and the single real-data experiment is simple and does not directly involve language modeling or prompts in the usual sense. Moreover, there is no empirical comparison against other DP training schemes (e.g., naive DP-SGD or DP ridge regression at the task level), which would better position NoisyHead in the DP literature.
> >
> > We have already added extra experiments comparing our method with the DP-SGD implementation in the Python package Opacus (Section 6.4)

---

> > > ### Comment · Reviewer_jHSy · 2026-02-16
> > >
> > > Many thanks for the response. They well answer my questions.

---

### Review · Reviewer_oayZ · 2025-12-24

**Summary Of Contributions:**

The paper studies the privacy–utility tradeoff of in-context learning (ICL) through the lens of differential privacy. Focusing on single-head linear attention for in-context linear regression, the paper proposed a differentially private training algorithm (NoisyHead) and provides theoretical bounds for cost of privacy that characterizing the impact of DP noise on ICL performance. The analysis highlights a noise accumulation phenomenon that necessitates early stopping under DP constraints, and the paper further presents robustness results against certain prompt perturbations, supported by synthetic and real-data experiments.

**Audience:**

Yes

**Audience Explanation:**

The topic is timely and relevant, and the paper contains nontrivial technical analysis.

**Claims And Evidence:**

Yes

**Claims Explanation:**

The proofs of main thereoms seem correct, and experimental results are generally consistent with the theoretical predictions.

**Requested Changes:**

1. While differential privacy is a central theme of the paper, its role and meaning in this work are not clearly articulated.
First, it is unclear what exactly is protected under DP in this setting. The paper defines neighboring datasets as differing in a single “prompt,” but does not explain what a prompt corresponds to in practice (e.g., a task, a document, or a sequence of tokens), nor why this is the appropriate unit of privacy in the context of ICL and large language models. The assumed threat model is also left implicit: what information is the adversary assumed to observe (model parameters, predictions, or something else)?
Second, the exposition of DP itself is very brief. Beyond stating the formal definition of DP, the paper does not provide sufficient discussion of how privacy is actually achieved by the proposed algorithm. In particular, the privacy guarantee relies on clipping, projection, Gaussian noise injection, and iterative updates, yet the paper does not clearly explain:
how the sensitivity is bounded at each step/how the clipping and projection operations contribute to this bound/how the noise scale is derived, and which composition or privacy accounting method is used across iterations.
As a result, the DP proof appears more as a high-level invocation of standard results (“by Gaussian mechanism and composition”) rather than a transparent and reproducible argument. Given that DP is a core claim of the paper, I believe a more detailed discussion is necessary.

2. The paper presents NoisyHead as a new differentially private training algorithm. However, from a privacy-mechanism perspective, NoisyHead closely resembles standard DP-GD approach, consisting of gradient clipping, Gaussian noise injection, and projection.
While the method leverages the structure of in-context learning by operating on prompt-level statistics, this distinction is not clearly articulated in the paper.
Clarifying this point is important for properly assessing the novelty of the contribution. At present, the paper does not clearly explain why NoisyHead is meaningfully different from applying a carefully designed DP-GD baseline under the same adjacency definition, nor how the sensitivity or noise requirements compare.

3. The Related Work section’s treatment of differential privacy appears superficial. While several classic and representative DP references are cited, the discussion largely takes the form of name-dropping rather than a structured review of the literature most relevant to this work.
In particular, the paper does not adequately survey or position itself within the recent literature on iterative DP optimization, including DP-GD, DP-SGD, and related variants (e.g., improvements in privacy accounting, iteration-dependent bounds, or utility–privacy tradeoffs). As a consequence, it is difficult for the reader to understand how the proposed method and analysis relate to existing results, or what is genuinely new from a DP perspective.
I strongly recommend expanding and organizing the DP-related work

---

> ### Author Response · Authors · 2026-02-12
> **Follow-up**
>
> Dear reviewer,
>
> Since the deadline of February 15 is approaching, kindly let us know if you have any further comments, which we will be happy to address.
>
> Best regards
> Authors

---

### Review · Reviewer_ZfeQ · 2026-02-01

**Summary Of Contributions:**

This paper investigates the theoretical intersection of Differential Privacy and In-Context Learning. The authors focus on a single-layer linear transformer trained to perform ICL on linear regression tasks.

1. Proposed NoisyHead, a DP-pretraining algorithm that applies the Gaussian mechanism (clipping and noise injection) to the attention weight
2. Deriving a "cost of privacy" bound that quantifies the utility gap in ICL performance under $(\epsilon, \delta)$-DP constraints.
3. Providing a theoretical link between DP-pretraining and adversarial robustness in the context of training prompts.
4. Simulations on synthetic linear regression tasks to verify the derived theoretical results on a single-layer linear transformer.

Strength:
1. Partially addresses an important and timely topic (privacy in ICL with transformer).
2. Translates the theoretical analysis of the privacy-utility trade-off from DP-SGD for the simplified transformer architecture.

Weaknesses:
1. Limited algorithmic novelty: The "NoisyHead" algorithm is essentially a direct application of DP-SGD to a ridge regression objective.
2. Over-simplified model: The use of a single-layer linear Transformer removes the non-linearities (e.g., Softmax) that define the complexity of noise propagation in real-world Transformers.
3. Theoretical overlap: Much of the "non-triviality" claimed appears to be a re-packaging of established DP-Linear Regression bounds within ICL terminology.

**Audience:**

Yes

**Audience Explanation:**

Despite the concerns regarding novelty, there could be a significant sub-community interested in the theoretical foundations of transformers and the formal guarantees of privacy. Translating DP-SGD results into the ICL framework provides a baseline for future researchers working on more complex, non-linear DP-ICL problems.

The authors also emphasize the $N = O(D^2)$, i.e. the high-dimensional regime. However, whether this is practical for transformers is questionable, as a practical transformer will contain billions/trillions of params. Although it doesn't hurt to state this regime as a special example, the authors should clarify the relevance of this regime to standard transformer application or shift focus to more realistic $N \ll D$ settings.

However, the interest would be significantly higher if the paper more clearly articulated what is unique to the transformer architecture that cannot be explained by simple linear regression theory.

**Broader Impact Concerns:**

Since this is a theory-heavy paper focused on enhancing privacy, there are no obvious red flags.

**Claims And Evidence:**

No

**Claims Explanation:**

1. While the mathematical derivations appear internally consistent, the claim that this work provides a "new" understanding of private ICL is hindered by the extreme simplification of the model. Moreover, the author provided no results on a larger scale of transformer training, which further limits the practical interest of the empirical ML community.

2. Equivalence to ridge Regression: In the single-layer linear case, the Transformer's objective function is mathematically equivalent to a ridge-regularized least squares problem. The NoisyHead algorithm (weight decay + clipping + Gaussian noise) is isomorphic to standard private ridge regression. The authors do not sufficiently distinguish their results from existing DP-optimization literature (e.g., [1]).


[1] Bolt-on Differential Privacy for Scalable Stochastic Gradient Descent-based Analytics

**Requested Changes:**

1. Distinction from DP-ridge regression: The authors must explicitly discuss how NoisyHead and its utility bounds differ from existing results in DP-linear regression. If the results are mathematically equivalent, this should be acknowledged, and the contribution should be reframed as a translation paper.

2. Discussion of non-linearity: The current analysis ignores the softmax activation. Since softmax is a non-contraction mapping, it significantly complicates noise sensitivity. The authors should at least provide a discussion or a small-scale experiment with softmax attention to demonstrate if the "linear" insights hold.

3. Compare the performance of NoisyHead against the baseline of calculating the ridge solution and adding noise at the end. This would help prove if iterative noise injection during training offers any benefit over simple noise injection

5. Acknowledgement of prior works: The following papers studied how Transformers learn statistical estimators; therefore are highly relevant to the theoretical framework used in this submission, but citations are currently missing:

[1] Training Dynamics of Multi-Head Softmax Attention for In-Context Learning: Emergence, Convergence, and Optimality
[2] In-context convergence of transformers
[3] In-Context Learning with Representations: Contextual Generalization of Trained Transformers
[4] One-Layer Transformer Provably Learns One-Nearest Neighbor In Context

---

> ### Author Response · Authors · 2026-02-11
> **Response to Reviewer ZfeQ**
>
> Thank you for your detailed comments. We respond to your comments on our claims, and requested changes in the following.
>
> >  **While the mathematical derivations appear internally consistent, the claim that this work provides a "new" understanding of private ICL is hindered by the extreme simplification of the model. Moreover, the author provided no results on a larger scale of transformer training, which further limits the practical interest of the empirical ML community.**
>
> We thank the reviewer for this remark. While it is true that we analyze a simplified architecture that intends only to address private pre-training of a linear attention head to ICL in linear regression problems, this choice is intentional and motivated by the fact that, even in this simplified setting, the trade-off between differential privacy (DP) and accuracy has remained poorly understood. Prior to our work, there was no known characterization of the privacy–utility frontier in this regime.
>
> Our contribution addresses this fundamental gap by providing, to our knowledge, the first near-optimal bound capturing the interplay between privacy guarantees and predictive performance in in-context learning with linear transformers. This result not only establishes a baseline for privacy-preserving learning in a controlled setting, but also serves as a theoretical stepping stone for studying more complex architectures and tasks.
>
> In a similar spirit, foundational works in the non-private setting -such as those by Garg et al. [‘What Can Transformers Learn In-Context? A Case Study of Simple Function Classes’, NeurIPS, 2022], Zhang et al. ([‘Trained Transformers Learn Linear Models In-Context’, JMLR, 2024, ‘In-Context Learning of a Linear Transformer Block’, NeurIPS, 2024]) - began by analyzing linear transformers and regression tasks to understand the capabilities of in-context learning before extending to richer scenarios. We believe that building a rigorous understanding in this stylized regime (as exemplified in many works including the works mentioned above) is a crucial prerequisite for meaningful generalizations to deep, non-linear transformers trained under privacy constraints.
>
> We admit that our work may not address all the concerns of the empirical ML community, where more complex architectures are usually studied; in this paper, our goal is to provide a theoretical understanding of the cost of privacy in pre-training attention heads. Therefore, for mathematical convenience, we start with a simpler set-up to accurately and rigorously characterize the privacy-utility trade-off. As is common in theoretically underdeveloped areas, such as the emerging theory of transformers, our objective is not to attain state-of-the-art benchmark accuracy, but rather to provide principled theoretical insight into mechanisms that may govern more complex empirical behavior. Indeed, a rigorous mathematical characterization of prediction risk in in-context learning (ICL) for general transformer architectures remains far from universal, and is poorly understood even in the non-private setting. We view this work as an initial step toward a broader theoretical understanding of privacy enforcement in modern attention-based models; however, extending such guarantees to more realistic architectures (including soft-max attention) is likely to require substantially more intricate and context-dependent mathematical analysis. In the conclusion, we have included studying the privacy-utility trade-off in more generic transformer architectures as a potential future direction.

---

> ### Author Response · Authors · 2026-02-11
> **Response to Reviewer ZfeQ (Ctd.)**
>
> >**Equivalence to ridge Regression: In the single-layer linear case, the Transformer's objective function is mathematically equivalent to a ridge-regularized least squares problem. The NoisyHead algorithm (weight decay + clipping + Gaussian noise) is isomorphic to standard private ridge regression. The authors do not sufficiently distinguish their results from existing DP-optimization literature (e.g., [1]).**
>
> We agree with the reviewer that we simplify our objective function to a ridge regression-based optimization framework. However, we respectfully disagree that our analysis is isomorphic to the DP ridge-regression commonly studied in the literature. In particular, for this problem, the features used in the objective to predict the response are not directly observed, but are rather constructed through nontrivial transformations of both the covariates and responses within each training prompt, mediated by the attention mechanism. Moreover, in the privacy-utility trade-off of a private ridge regression, the determining factors of the prediction accuracy are $N$ (sample size) and $D$ (dimension). In contrast, for the ICL setting, we have three determining parameters: the prompt length $L$, the dimension of each token $D$, and the total number of training prompts $N$, introducing interesting interplay in the statistical rates of prediction, which are significantly different from the standard analysis of DP-ridge regression, as well as previous works. We have included a remark in this regard in Remark 4.3 in the updated manuscript, following Proposition 4.1.
>
>  However, the optimization objective is solved using a modified version of DP-GD; we have elaborated on the nuances and differences in that regard in a discussion in Pages 3 \& 4, in red. For the convenience of the reviewer, we reproduce it in the following.
>
>   > It is worthwhile to mention that some recent works have also explored differentially private fine-tuning on complicated models such as GPT-2 or ViT \citep{yu2023dptraining, delvingding, oh2024privacypreserving}, which mainly apply DP-SGD to fine-tune such models. \revsn{It is worth mentioning that \fancyname\, structurally, resembles a carefully designed DP-GD pipeline that utilizes the structure of the specific in-context learning problem and the model architecture to carefully calibrate the amount of noise infused in the pipeline. While both our pipeline and the standard DP-SGD pipeline involve clipping, projection, and noise injection, they differ significantly in structure and intent. In particular, if applied naively, DP-SGD would treat the entire sequence of $L$ context-label pairs as private and apply noise accordingly. This would cause the privacy cost to compound across the entire episode, leading to substantial degradation in accuracy. In contrast, \fancyname\ leverages the structure of in-context learning by injecting noise only at the level of the final gradient, using carefully designed clipped sufficient statistics that summarize each episode in a privacy-preserving way. This includes clipping of the response variables $y_{k,i}$ before computing sufficient statistics-a step not typically used in standard DP-SGD but essential for both privacy guarantees and statistical accuracy, particularly with long contexts. Furthermore, our clipping constants are derived using the model assumptions on the prompt level features and responses using appropriate concentration inequalities. While this derivation is inspired by analogous analyses in linear regression settings \cite{tcai-cost-of-privacy}, the corresponding calculations for the in-context learning problem are significantly more involved and, to the best of our knowledge, have not been explored previously. Finally, DP-SGD is mainly designed for classical supervised learning, where model parameters are updated with each example. In contrast, \fancyname\ generalizes across tasks presented at test time and injects noise only at the head level, using aggregated, privatized task-level statistics. Finally, despite showing promising empirical performance, DP-SGD-based pre-training of attention modules often lacks theoretical guidance on how privacy cost scales with model structure, sample size, or representation geometry. In contrast, we precisely characterize the amount of extra price paid in the prediction loss due to enforcing privacy in the training pipeline at the user-specified threshold. Our theoretical insights provide a rigorous foundation that could inform the design of future DP algorithms for large-scale models with a quantifiable characterization of the cost of privacy in the in-context learning performance.

---

> ### Author Response · Authors · 2026-02-12
> **Response to Reviewer ZfeQ (Ctd.)**
>
> > **Distinction from DP-ridge regression:** *The authors must explicitly discuss how NoisyHead and its utility bounds differ from existing results in DP-linear regression. If the results are mathematically equivalent, this should be acknowledged, and the contribution should be reframed as a translation paper.*
>
> Addressed above.
>
> > **Discussion of non-linearity:** *The current analysis ignores the softmax activation. Since softmax is a non-contraction mapping, it significantly complicates noise sensitivity. The authors should at least provide a discussion or a small-scale experiment with softmax attention to demonstrate if the "linear" insights hold.*
>
> Thanks for pointing out the excellent references on in-context learning using soft-max attention. We were not aware of this literature, and in the revised manuscript, we have included a discussion on these papers (in Related Section and Conclusion). Although extension of our algorithm is possible, and the results could be extended in the non-linear setting by combining the ideas of the framework, as ``How do nonlinear transformers learn and generalize in in-context learning? (ICML 2024)"; however, such an analysis will not be as clean as in the linear case. We believe that the scaling laws might still hold. But it needs extensive training and further non-trivial theoretical justification, which we aim to investigate in future work. We have added a discussion in the conclusion section about this, and the discussion reads as follows:
>
> *A recent line of work has analyzed in-context learning performance of in architectures involving soft-max attention Chen et al., 2024; Huang et al., 2023; Yang et al., 2024; Li et al., 2024a). It is an interesting future direction to extend our methods to such architectures and characterize the privacy-utility trade-off.*

---

> ### Author Response · Authors · 2026-02-12
> **Response to Reviewer ZfeQ (Ctd.)**
>
> > **Compare the performance of NoisyHead against the baseline of calculating the ridge solution and adding noise at the end. This would help prove if iterative noise injection during training offers any benefit over simple noise injection.**
>
> Thanks for this suggestion. Based on your suggestion, we have included a discussion along with some simulation results in the appendix, where we have compared NoisyHead with computing the ridge regression-based solution to (2.6) and adding calibrated Gaussian noise to the final solution to produce a differentially private weight matrix $\hat{\Gamma}^{\text{DP}}$. The noise injection in the final step (ridge regression) provides better prediction error compared to NoisyHead in finite samples, though we expect that the scaling laws will remain similar compared to the NoisyHead-based estimate. The worsened performance in NoisyHead is due to early stopping in an iterative noise injection scheme, required to ensure differential privacy. However, this mechanism of injecting noise directly onto the actual estimate is critically tuned to the linear attention framework, and the insights gained from such a mechanism do not transfer directly to generic architectures such as those based on softmax-based attention. In fact, for more sophisticated architecture, such a mathematically quantifiable solution is usually not available. In contrast, one can always train any reasonable architecture using DP-GD-type optimization as proposed in NoisyHead, which makes controlling the sensitivity of the gradient step much more feasible, even if mathematically non-trivial.  Therefore, to provide a baseline for future research on the scaling laws of prediction error in attention-based architecture, we stick to DP-GD type iterative training proposed in NoisyHead.
>
> On the insistence of the reviewer, we explain the alternative ridge-regression-based mechanism that you proposed in a remark following Theorem 3.2 (Remark 3.1) and explain the details of producing a privatized ridge regression-based estimate in the Appendix B. We also provide a simulation study to compare the performance of NoisyHead and the ridge-regression-based estimate in the low-dimensional setting. To ensure a fair comparison, we computed the ridge estimate as in equation (2.7) in the current manuscript, based on clipped matrices and response, as follows:
> $$
> \mathrm{vec}(\Gamma^\star; E_1,\ldots, E_N) = ( \lambda N I + \sum_{k=1}^N \mathrm{vec}(\tilde{Z}_k) \mathrm{vec}(\tilde{Z}_k)^\top )^{-1} \sum\_{k=1}^N \mathrm{clip}\_{\mathcal{C}}(y\_{k, L+1}) \mathrm{vec}(\tilde{Z}_k)
> $$
>
> where $\mathcal{C}, G$ and $R$ are as in Theorem 4.1. The clipped and projected inputs ensure stability, and control the sensitivity of the ridge estimator. Finally, to compute the corresponding prediction risk, we further project $\Gamma^\star$ to a Frobenius ball of radius $CG/ (\lambda N)$. The Frobenius sensitivity of the projected estimator to the perturbation of one input prompt is controlled at $2R$, and therefore by Gaussian mechanism, we add an i.i.d. Gaussian noise with variance $8\frac{C^2G^2}{\varepsilon^2 \lambda^2 N^2}\log \frac{1.25}{\delta}$. This ensures the final estimate is $(\varepsilon, \delta)$-differentially private. The following table shows this estimator has a significantly higher risk compared to the Noisyhead estimate.
>
> | $N$ | $\varepsilon$ | Noisyhead | DP-ridge |
> |-----|--------------|------------|----------|
> | 2000 | 0.2 | 0.1302 | $5.86\times 10^{-5}$ |
> | 2000 | 0.4 | 0.1305 | $1.45\times 10^{-5}$ |
> | 3000 | 0.2 | 0.07280 | $9.73\times 10^{-6}$ |
> | 3000 | 0.4 | 0.03517 | $2.44\times 10^{-6}$ |
> | 4000 | 0.2 | 0.02597 | $2.71\times 10^{-6}$ |
> | 4000 | 0.4 | 0.00652 | $6.75\times 10^{-7}$ |
>
> > **Acknowledgement of prior works:** *The following papers studied how Transformers learn statistical estimators; therefore are highly relevant to the theoretical framework used in this submission, but citations are currently missing:*
>
> We have added the references in Related Literature Section 1.2 (colored in red). We apologize for the oversight and appreciate the reviewer for pointing out these references.

---

### Decision · Action_Editor_DE3b · 2026-06-04

**Recommendation:** Accept as is

**Additional Comments:**

This paper studies differential privacy for in-context learning in simplified setting with a linear attention head trained for in-context linear regression. The paper proposes a private pretraining procedure, NoisyHead, and provides a theoretical analysis of the resulting privacy-utility tradeoff, including low- and high-dimensional regimes, the role of early stopping under iterative noise injection, and robustness to adversarial perturbations of training prompts.

The main strength of the submission is that it appears technically solid within its stated scope. The reviewers found that the theoretical analysis is sound and the topic is timely, and I agree that the paper makes a meaningful contribution as a theory paper at the intersection of privacy and in-context learning.

The main limitation is scope. The model remains highly stylized, the experiments are relatively limited, and the paper does not substantially close the gap to realistic softmax-based transformers or large-scale language-model training. These concerns, however, primarily limit the breadth of the claims rather than undermine the correctness of the technical contribution.

On balance, I this the paper is a solid work, and I therefore recommend acceptance.

**Audience:**

Yes

**Audience Explanation:**

The paper addresses the intersection of two topics of clear interest to the TMLR community: theoretical understanding of in-context learning and differential privacy in modern learning systems.

**Claims And Evidence:**

Yes

**Claims Explanation:**

The main theoretical claims are supported by formal statements and detailed proofs. The empirical results are relatively limited in scope, but they are still generally well-aligned with the theory and support the paper's central qualitative conclusions.